# Pervasive tandem duplications and convergent evolution shape coral genomes

Benjamin Noel[1,2†], France Denoeud[1,2†], Alice Rouan[3,4], Carol Buitrago-López[5], Laura Capasso[4,6,7], Julie Poulain[1,2], Emilie Boissin[8], Mélanie Pousse[3,4], Corinne Da Silva[1,2], Arnaud Couloux[1,2], Eric Armstrong[1,2], Quentin Carradec[1,2], Corinne Cruaud[2,9], Karine Labadie[2,9], Julie Lê-Hoang[1,2], Sylvie Tambutté[4,6], Valérie Barbe[1,2], Clémentine Moulin[2,10], Guillaume Bourdin[11], Guillaume Iwankow[8], Sarah Romac[12], Sylvain Agostini[13], Bernard Banaigs[8], Emmanuel Boss[11], Chris Bowler[2,14], Colomban de Vargas[2,12], Eric Douville[15], J. Michel Flores[16], Didier Forcioli[3,4], Paola Furla[3,4], Pierre E. Galand[2,17], Fabien Lombard[2,18,19], Stéphane Pesant[20], Stéphanie Reynaud[4,6], Matthew B. Sullivan[21], Shinichi Sunagawa[22], Olivier P. Thomas[23], Romain Troublé[2,10], Rebecca Vega Thurber[24], Denis Allemand[4,6], Serge Planes[2,8], Eric Gilson[3,4,25], Didier Zoccola[4,6], Patrick Wincker[1,2], Christian R. Voolstra[5] and Jean-Marc Aury[1,2*]

†Benjamin Noel and France Denoeud have equal contribution.

*Correspondence:
jmaury@genoscope.cns.fr

[1] Génomique Métabolique, Genoscope, Institut François Jacob, CEA, CNRS, Univ Evry, Université Paris-Saclay, Evry 91057, France
Full list of author information is available at the end of the article

## Abstract

**Background:** Over the last decade, several coral genomes have been sequenced allowing a better understanding of these symbiotic organisms threatened by climate change. Scleractinian corals are reef builders and are central to coral reef ecosystems, providing habitat to a great diversity of species.

**Results:** In the frame of the Tara Pacific expedition, we assemble two coral genomes, *Porites lobata* and *Pocillopora* cf. *effusa,* with vastly improved contiguity that allows us to study the functional organization of these genomes. We annotate their gene catalog and report a relatively higher gene number than that found in other public coral genome sequences, 43,000 and 32,000 genes, respectively. This finding is explained by a high number of tandemly duplicated genes, accounting for almost a third of the predicted genes. We show that these duplicated genes originate from multiple and distinct duplication events throughout the coral lineage. They contribute to the amplification of gene families, mostly related to the immune system and disease resistance, which we suggest to be functionally linked to coral host resilience.

**Conclusions:** At large, we show the importance of duplicated genes to inform the biology of reef-building corals and provide novel avenues to understand and screen for differences in stress resilience.

## Introduction

Coral reefs are one of the most diverse ecosystems on the planet. Although covering less than 0.2% of the ocean floor, coral reefs are home to over 25% of all described marine species [1, 2]. Coral reefs also provide coastal protection and services to human societies. They support the livelihoods of millions of people through fishing or tourism [3]. Reef-building corals are the foundation species of coral reefs with critical roles in their function and maintenance. At the heart of this complex ecosystem, coral holobionts are meta-organisms composed of three main components: the coral host (cnidaria), photosynthetic Symbiodiniaceae (dinoflagellates), and associated prokaryotes, among other organismal entities [4, 5].

For several decades now, coral reefs have been declining, impacted by global ocean warming, besides local anthropogenic impacts [6–9]. This temperature increase disrupts the coral and Symbiodiniaceae symbiosis leading to massive coral bleaching and mortality [10]. In addition, ocean acidification due to the increased levels of atmospheric $CO_2$ concentration reduces the ability of coral to produce its calcium carbonate skeleton and lowers its resilience [11, 12]. Recently, the Intergovernmental Panel on Climate Change (IPCC) reported a projected decrease of 70 to 90% of the coral reefs coverage even if global warming is constrained to 1.5℃ [13]. This will drastically affect reef ecosystems and spurs incentives to develop mitigation strategies besides the curbing of $CO_2$ emissions [14–16].

Marine sessile species have a wide range of lifespans, ranging from weeks to thousands of years for some deep-sea corals and sponges [17]. Habitat depth appears to play a key role: indeed, at greater depths, living organisms are protected from issues that affect species in shallower waters, such as climatic temperature changes, climate events, and most importantly human activity [18]. Despite their fragility, corals are resilient organisms and several species have colonies with an extreme longevity, of the order of hundreds or thousands of years [17, 19]. This could be seen as a paradox for those sessile species that cannot evade external threats and environmental changes. However, corals form colonies of multiple genetically identical and independent individuals, called polyps, and clonal organisms can escape age-related deterioration [20, 21]. In addition, the colony can remain functional over time, even though parts may die.

The genome of *Acropora digitifera* was published 10 years ago and was the first scleractinian coral genome available [22]. This first opportunity to uncover the architecture of a coral genome revealed a general absence of gene transfer with the endosymbiont, despite their long evolutionary relationship, a contradictory result with a more recent study [23]. It also highlighted the capacity to synthesize ultraviolet-protective compounds, the presence of genes with putative roles in calcification and a complex innate immunity repertoire with a putative role in the coral-Symbiodiniaceae symbiosis. With the further development of short-read sequencing technologies, a large number of coral genomes have been sequenced over the past decade [24–31]. The analysis of short-read assemblies from two corals of the highly diverged complex and robust clade [32] suggests, on the one hand, that many gene families exhibit expansion in corals (in particular genes having a role in innate immunity), and on the other hand that these gene family expansions have occurred independently in complex and robust corals [24]. Tandem organization of these expanded gene families was suggested, as some amplified genes of

a given family are localized on the same scaffolds [26, 33]. Coral genomes are diploid and often highly heterozygous, which represents a major difficulty in generating high-quality genomes [31, 34]. Owing to the circumstance of short-read sequencing, many available coral genomes exhibit low contiguity and incomplete assemblies. Even though it has been generally accepted that short-read assemblies are exhaustive for genes, repetitive regions are generally underrepresented [25, 35], and in particular tandemly duplicated genes are a special case of repetitive regions which may be missed [35, 36].

Here we report high-quality genome assemblies of two globally prevalent corals sampled through the Tara Pacific expedition [37]: the complex coral *Porites lobata* and the robust coral *Pocillopora* cf. *effusa*, based on long reads generated using the Oxford Nanopore technology (ONT). In addition, we sequenced a second genome of a morphologically similar *Porites* (*Porites evermanni*) with divergent stress susceptibility using short reads [38]. On the basis of these three genomes and other available cnidarian genomes, we carried out a broad comparative analysis. Our results expose the vast presence of duplicated gene families in both coral genomes mapping to functions associated with the innate immune system, which escaped previous analyses based on fragmented and incomplete genomes assemblies due to sequencing method constraints. We posit that these tandem duplications shape current coral genomes and contribute to the longevity of these organisms, especially in *Porites lobata* where colonies have been described that are over 1000 years old [39].

## Results

### Coral genome sequence assemblies and gene catalogs

The *P. lobata* and *P.* cf. *effusa* genomes were generated using a combination of ONT long reads and Illumina short reads (Additional file 1: Table S1 and Additional file 1: Table S2). Using kmer distributions, *P. lobata* and *P.* cf. *effusa* genome sizes were estimated to be 543 Mb and 315 Mb respectively, and a high level of heterozygosity was detected, 2.3 and 1.14% respectively (Additional file 1: Figure S1). As cumulative size of the two genome assemblies was almost twice as large as expected, and subsequent analyses revealed the presence of allelic duplications, Haplomerger2 [40] was used on both assemblies to generate an assembly of reference and alternative haplotypes (Additional file 1: Figure S2, Additional file 1: Figure S3, Additional file 1: Table S3 and Additional file 1: Table S5). The reference haploid assemblies, with cumulative sizes of 646 Mb for *P. lobata* and 347 Mb for *P.* cf. *effusa*, contained 1098 contigs and 252 contigs with N50 of 2.15 and 4.7 Mb, respectively (Table 1). In addition, we sequenced the genome of *Porites evermanni* using short-read technology. Although more fragmented, this assembly has been used to perform comparative genomic analyses. Despite the fact that a large fraction of the repetitive elements is still unknown in the here-sequenced coral genomes (Additional file 1: Table S6), DNA transposons were detected as the most abundant in the *P. lobata* genome (representing 17.4%), and in contrast, the most abundant repeat type was retroelements in the *P.* cf. *effusa* genome (10.5%). We annotated the three genomes using transcriptomic data (for *P. lobata* and *P.* cf. *effusa*) and 25 cnidarian proteomes, resulting in 42,872, 40,389 and 32,095 predicted genes for *P. lobata*, *P. evermanni*, and *P.* cf. *effusa* respectively (Table 1). During the annotation process, we identified alignments of known proteins and transcripts that span large genomic regions

**Table 1** Statistics of the three coral genome assemblies from this study compared to representative existing genomes of the same clades

| | Complex | | | | Robust | | |
|---|---|---|---|---|---|---|---|
| | Porites lobata | Porites evermanni | Porites lutea | Acropora millepora | Pocillopora cf. effusa | Pocillopora verrucosa | Pocillopora damicornis |
| Publication | This study | This study | Robbins et al | Fuller et al | This study | Buitrago-López et al | Cunning et al |
| Estimated genome size | 543 Mb | 497 Mb | 552 Mb[a] | ? | 315 Mb | 407 Mb[a] | 262 Mb[a] |
| # contigs\|scaffolds | 1098 | 8186 | 2975 | 854 | 252 | 18,268 | 4393 |
| Cumulative size | 646,152,978 | 603,805,388 | 552,020,673 | 475,381,253 | 347,233,126 | 380,505,698 | 234,335,492 |
| N50 (L50) | 2,154,615 (84) | 171,385 (935) | 660,708 (242) | 19.8 Mb (9) | 4,753,879 (23) | 333,696 (326) | 326,133 (198) |
| Max size | 8,615,247 | 1,802,771 | 3,122,227 | 39,361,238 | 11,895,822 | 2,095,917 | 2,168,405 |
| # of N's | 0 (0%) | 40,756,223 (6.75%) | 48,123,166 (8.72%) | 37,012 (0.01%) | 0 (0%) | 510,035 (0.13%) | 8,607,682 (3.67%) |
| # contigs | 1098 | 32,888 | 47,330 | 1234 | 252 | 54,131 | 53,036 |
| N50 (L50) | 2,154,615 (84) | 33,681 (4,563) | 19,557 (7,534) | 1,091,365 (129) | 4,753,879 (23) | 23,429 (3,851) | 25,941 (2,282) |
| Repeat coverage (% of assembly) | 51.28 | 42.26 | 42.36 | ? | 36.67 | 38.44 | 20.36 |
| # number of genes | 42,872 | 40,389 | 31,126 | 28,188 | 32,095 | 27,439 | 26,077 |
| Genes density (genes/Mb) | 66.4 | 66.7 | 56.4 | 59.3 | 92.5 | 72 | 111.4 |
| % BUSCO (compl.; frag.; miss.) eukaryota odb10 $N=255$ genes | 97.7; 1.2; 1.1 | 94.5; 3.9; 1.6 | 92.2; 4.3; 3.5 | 73.7; 16.5; 9.8 | 98.4; 0.4; 1.2 | 90.2; 5.1; 4.7 | 86.3; 9.0; 4.7 |

[a] Data from publications

(Additional file 1: Figure S10) and further investigations indicated that these regions contain tandemly duplicated genes (TDG). These duplicated genes are generally difficult to assemble and to predict accurately. Here we developed a new annotation process to systematically improve the annotation of these TDG. Finally, gene completeness was estimated using BUSCO and was 97.7, 98.4 and 94.5%, respectively (Table 1).

### Telomeric sequences

Telomeres are composed of short repeated DNA sequences located at the end of linear eukaryotic chromosomes. The telomeric repeat motif TTAGGG is highly conserved among metazoans [41]. This type of sequence can also be found within the chromosome and is therefore called interstitial telomeric sequence (ITS). We identified ITSs in our three genome assemblies and the previously sequenced *Stylophora pistillata* [24], along with a low proportion of contigs with telomeric repeats at their ends (5 and 3 for *P.* cf. *effusa* and *P. lobata*, respectively), suggesting the absence of contigs representing complete chromosomes and the quasi-absence of terminal chromosome fragments. This absence of telomeric repeats at contig ends may be the consequence of a technical issue during the basecalling of nanopore data [42]. Strikingly, we noticed in the three *Porites* species (*P. lobata*, *P. evermanni* and *P. lutea*) the presence of a 188-nt length satellite DNA sequence containing a palindromic telomeric sequence (Additional file 1: Figure S11). These satellites are found tandemly repeated in intergenic regions. Attempts to search for this sequence failed outside the *Porites* genus.

## Comparison with available coral genomes

To date, only three other scleractinian genomes, i.e., *Montipora capitata, Acropora millepora,* and *Acropora tenuis,* have been assembled using long-read sequencing technologies [28, 43, 44], and the genome sequences of *P. lobata* and *P.* cf. *effusa* we have generated are the most contiguous and complete coral genome assemblies so far (Fig. 1). Likewise, we observed that several genomes have a high number of duplicated BUSCO

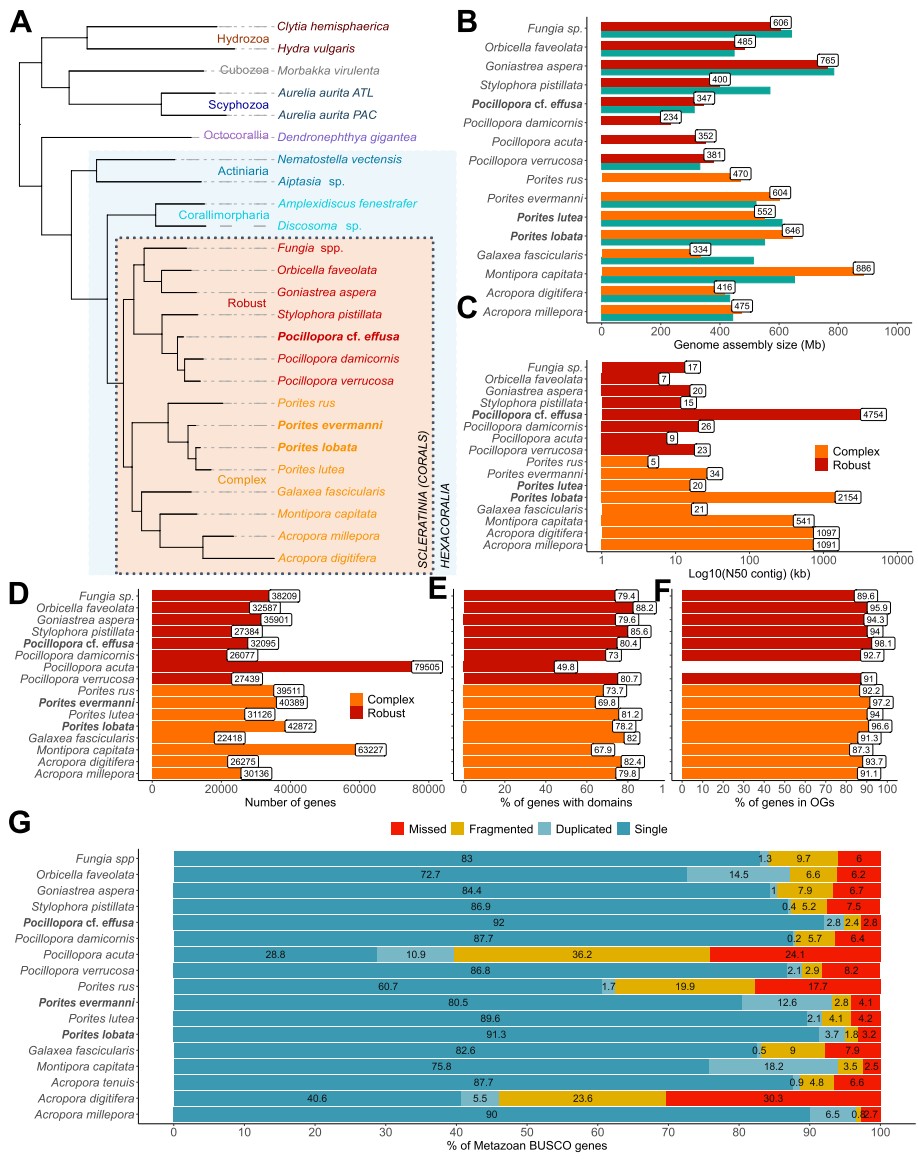

**Fig. 1** Comparison of available coral genomes. Species from the complex clade are in orange, species from the robust clase are in red and the three genomes described in this study are in bold. **A** Rooted species tree of 25 cnidarian species based on OrthoFinder. **B** Genome assembly sizes are in megabases, green bars indicate the estimated genome size based on kmers calculated from short reads when available. **C** Contig N50 values in kilobases (log scale). **D** Number of annotated genes. **E** Proportion of genes containing a functional domain. **F** Proportion of genes in orthogroups (OG) that contain at least two different species. **G** BUSCO scores computed with the Metazoan gene set (*N* = 954 genes). Numbers in the blue bar represent the proportion of complete and single-copy genes in each gene catalog. NB: see Table S7 for information on assembly/annotation versions used

genes, indicating that they still contain allelic duplications, potentially due to the afore-mentioned high levels of heterozygosity (Fig. 1G). Coupled with a fragmented assembly, these remaining duplications are detrimental for subsequent analysis, as it is then complicated to differentiate true duplicates from allelic copies of a given gene. In our assemblies of *P. lobata* and *P.* cf. *effusa*, BUSCO and KAT analyses showed a reduction of the allelic duplications which suggests that the two allelic versions were successfully separated as much as possible with currently available tools (Fig. 1G, Additional file 1: Table S3, Additional file 1: Table S5, Additional file 1: Figure S2, Additional file 1: Figure S3 and Additional file 1: Figure S4). To further compare coral genomes, orthologous relationships within 25 cnidarian species were identified (Additional file 1: Table S7). As expected, conservation between orthologous genes inside the *Porites* and *Pocillopora* genera was high. Surprisingly, however, the conservation of orthologous genes between *P. lobata* and other robust corals was as low as the conservation to other complex corals that are at least 245 mya apart [45] (Additional file 1: Figure S12). Notably, as an initial matter for debate, the classification of coral species into two evolutionary divergent clades (complex and robust) is recognized as real [26, 32] and confirmed in our analyses (Fig. 1A). As few morphological or biological criteria resolve the two groups [26], we suggest here that guiding the analyses by splitting into robust and complex clades does not always make sense, and alternative grouping could be sometimes more relevant to compare coral genomes. In addition, these orthologous relationships allowed us to examine the conservation of gene order within corals. As already reported, synteny across complex and robust coral lineages is highly conserved [26, 33] (Fig. 2). With their higher contiguity, the two long-read assemblies better resolve the macro- and micro-synteny within each of the two lineages. Interestingly, the synteny between complex and robust corals, which was previously described as conserved [26], is not conserved at the scale of large genomic regions. Indeed, fragmented assemblies give only a partial insight into the synteny between organisms. Here, we observed only a conservation at the micro-synteny level between *Porites* and *Pocillopora*, and despite the 245 Mya that separate these species, the syntenic blocks nevertheless cover at least 75% of both genomes (Fig. 2 and Additional file 1: Table S6). In comparison, only 40% of the assemblies are covered if comparing the two short-read assemblies of *P. lutea* and *P. verrucosa*, showing the shortcomings associated with analyzing fragmented genomes.

### Tandemly duplicated genes

Tandem duplications are an important mechanism in the evolution of eukaryotic genomes, notably allowing the creation of unconstrained genes that can lead to new functions [46–48], in particular for genes clustered into gene families [49, 50]. In our two high-quality assemblies of *P. lobata* and *P.* cf. *effusa*, we predicted more genes than in other coral species of the same genus (Fig. 1D) with a proportion containing conserved domains comparable to other corals (78 and 80% respectively, Fig. 1E). This higher number of genes can be related to a high number of tandemly duplicated genes (TDG). Indeed, we detected TDG in the available Cnidaria genome assemblies and annotations and found a high proportion of TDG in *P. lobata* and *P.* cf. *effusa*, 29.9% (12,818 genes) and 32.6% (10,449 genes) of their respective gene catalog. In comparison, the proportion of TDG is lower in short-read assemblies (Fig. 3A), except for *Orbicella faveolata* which

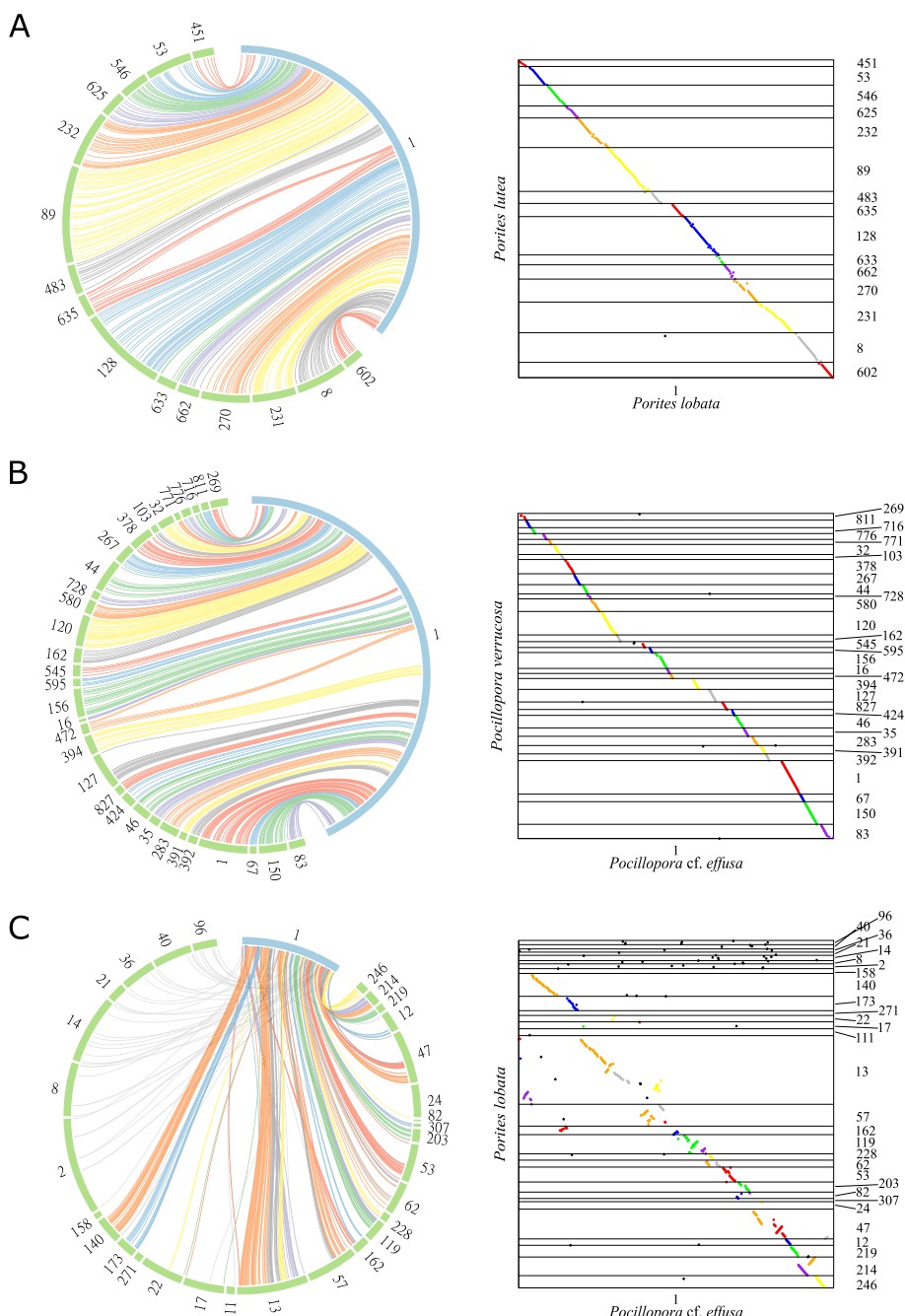

**Fig. 2** Coral synteny. Circular (left) and dotplot (right) representations of the synteny between the longest contigs. Each colored link represents linkage between two orthologous genes which are in a syntenic cluster. Colors of links represent syntenic clusters. Gray links connect orthologous genes that are not syntenic. Dotplots display only regions of contigs that contain orthologous genes. **A** Synteny between the longest contig of *P. lobata* (blue) and its syntenic scaffolds in *P. lutea* (green). **B** Synteny between the longest contig of *P.* cf. *effusa* (blue) and its syntenic scaffolds in *P. verrucosa* (green). **C** Synteny between the longest contig of *P.* cf. *effusa* (blue) and its syntenic contigs in *P. lobata* (green)

also displays a high proportion of allelic duplications making TDG detection confusing (Fig. 1G). Clusters of TDG are scattered on all contigs (Additional file 1: Figure S13) and contain on average two genes in both coral genomes, but some clusters contain more

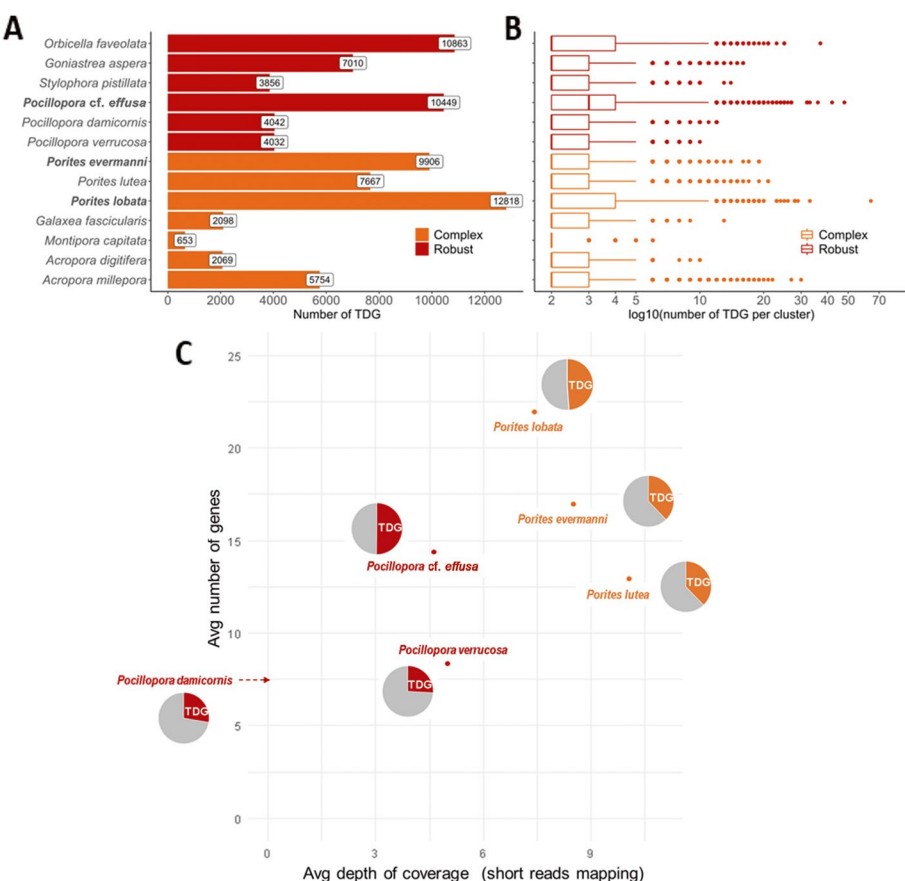

**Fig. 3** Quantification of tandemly duplicated genes (TDG) in coral genomes. **A** Number of TDG for each species. **B** Distribution of the number of genes per TDG cluster. **C** For 499 gene families (orthogroups with ≥ 10 genes in *P.* cf. *effusa* or *P. lobata*), the number of genes in *Pocillopora* and *Porites* species is compared to the normalized depth of mapping of short reads on OG consensus (i.e., estimated gene copy number based on mapping of short reads). Pie charts represent the proportion of TDG genes in each species. For *Pocillopora damicornis*, no value of depth was computed since we were not able to identify a set of Illumina short reads to download

genes up to a maximum of 64 genes for *P. lobata* and 48 genes for *P.* cf. *effusa* (Fig. 3B). In general, we found larger clusters of TDG in genome assemblies with higher N50, which is expected as larger genomic sequences contain more candidate genes.

One could hypothesize that these TDG arise from assembly biases and are the result of uncaptured allelic duplications or false joins, especially during the Haplomerger stage. We compared our assemblies with the one produced by Purge Dups [51], another tool dedicated to remove haplotypic duplications, and demonstrated that, on the two assemblies, Haplomerger2 has reduced heterozygous duplication and maintained completeness while increasing assembly contiguity (Additional file 1: Table S3 and Additional file 1: Figure S4). In addition, we annotated the alternative haplotype assembly of the two species (Additional file 1: Table S4). In both cases, the number of TDG identified in the reference and alternative haplotypes is similar (3692 and 3599 respectively for *P. lobata*, 2741 and 2951 for *P.* cf. *effusa*) as well as the number of genes in TDG clusters (Additional file 1: Figure S5). There is a high concordance between TDG clusters in both haplotypes: 75 and 88% of TDG clusters in *P. lobata* and *P.* cf. *effusa* from the reference

haplotype have all their best reciprocal hit on the alternative haplotype in one single TDG cluster (exemple in Additional file 1: Figure S9A). Not surprisingly, synonymous substitution rates (Ks) are very distinct between TDG and allelic pairs (Fig. 5B), suggesting that the majority of detected TDG, often poorly conserved, do not correspond to artifacts where both haplotypes were assembled together. Moreover, respectively 70 and 91% of adjacent pairs of duplicated genes were validated by at least one Nanopore read in *P. lobata* and *P.* cf. *effusa* (Additional file 1: Table S9, Additional file 1: Figure S6 and Additional file 1: Figure S7) and for respectively 45 and 67% of TDG clusters, we were able to identify Nanopore reads that span the whole cluster, confirming the organization of these duplicated genes (Additional file 1: Table S9, Additional file 1: Figure S6 and Additional file 1: Figure S8). Additional file 1: Figure S9B shows an example where each haplotype assembly is validated by at least one Nanopore read.

The fact that other coral genomes have a lower proportion of TDG than *P. lobata* and *P.* cf. *effusa* was surprising, and we investigated whether this difference could be due to biases in the genome assembly or gene prediction workflows. To be independent from such biases, the number of members in gene families was estimated using short-read-based data and conserved genes. Orthologous genes computed within the 25 Cnidaria species (Additional file 1: Table S7) were grouped into orthogroups (OG) and a consensus sequence was built for each OG. The number of gene copies per OG was estimated for each species by aligning short-read sequencing data of the corresponding species to each OG consensus. Normalization was performed using 705 coral-specific and single-copy genes. The estimated gene copy number based on mapping of short reads is similar among all *Porites* and all *Pocillopora* species, whereas the number of annotated gene copies is higher for *P. lobata* as well as for *P.* cf. *effusa*. We found that *Porites* gene catalogs of short-read assemblies (*P. lutea* and *P. evermanni*) lack a high number of copies when compared to *P. lobata*, and the same trend was observed when comparing *Pocillopora* short-read assemblies with *P.* cf. *effusa* (Fig. 3C). Genome assemblies based on short reads thus appear to lack a substantial number of gene copies, particularly in TDG clusters. Indeed, *P. lobata* and *P.* cf. *effusa* have a higher number of genes but also a higher proportion of genes linked to an OG (respectively 96.6 and 98.1% of the genes are in an OG composed of genes from at least two different species, Fig. 1F), suggesting that their gene annotation is a better representation of the gene catalog of coral genomes. Interestingly, the *P. evermanni* assembly is also based on short reads but our improved gene prediction method appears to exonerate the number of missing gene copies (Fig. 3C).

### Amplified gene families in corals

To assess whether TDG contributed to gene family expansions, we searched for orthogroups (OG) with significant gene number differences between corals and sea anemones. We identified 192 OG that were expanded in corals (Additional file 2) in comparison to only 28 OG in sea anemones (Fig. 4). Most of the expanded gene families contained a high ratio of TDG (Fig. 4C), which suggests that tandem duplication is an important mechanism for gene family amplification in corals. The functions of amplified OG, based on InterProscan domain identification and blastP searches, correspond in vast majority to transmembrane receptors, cell adhesion, and extracellular signal transduction. The most abundant domain is G-protein-coupled receptor, rhodopsin-like (GPCR), that

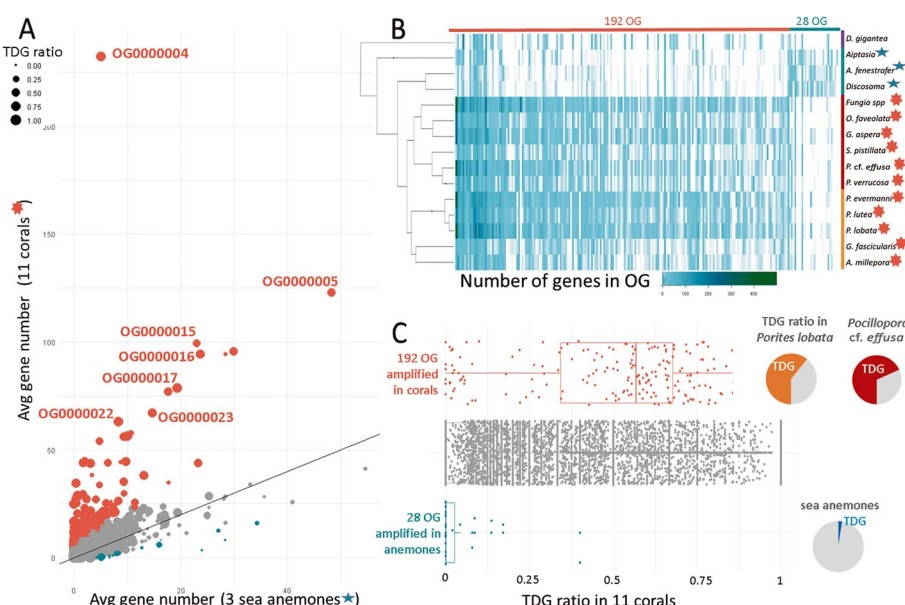

**Fig. 4** Amplified gene families in corals vs sea anemones. **A** Average number of gene copies in corals vs sea anemones. Orthogroups colored in orange have significantly more gene copies in corals compared to sea anemones and orthogroups colored in blue have significantly less gene copies in corals compared to sea anemones (binomial test, adjusted *p*-value < 0.001). Dot sizes correspond to the ratio of TDG for each OG in 11 coral genomes. **B** Heatmap of gene copy numbers in 15 species for 192 OG amplified in corals and 28 OG amplified in sea anemones. The phylogenetic tree is the output of the OrthoFinder software. **C** Proportion of TDG in 192 OG amplified in corals (orange), 28 amplified in sea anemones (blue), and not amplified OG (gray). The pie charts represent the proportion of TDG among the OG amplified in corals or sea anemones, in *Porites lobata* (orange), *Pocillopora* cf. *effusa* (red) and in sea anemones (blue)

corresponds to 34/192 OG amplified in corals. Among the receptors that were identified, some are involved in innate immunity and possibly coral/Symbiodiniaceae symbiotic relationships [52]. As previously reported in other coral species [24, 25], we observe a high heterogeneity of copy numbers in gene families among coral genera: each coral genus displays specific gene family expansions, with similar profiles among *Porites* species and among *Pocillopora* species (Additional file 2 and Fig. 4B). Additionally, we detect more pronounced amplifications in five genome species, i.e., *Porites lobata*, *Porites evermanni*, *Pocillopora* cf. *effusa*, *Goniastrea aspera*, and *Fungia* sp. However, the lack of high-quality assemblies and annotations for some of the genomes did not allow us to ascertain the biological significance of these observations. These results confirm on a larger range of species and at a higher scale that although the same functional categories (extracellular sensing, cell adhesion, signalling pathways) are amplified in all corals, individual amplified gene families diverge among genera, corroborating previous notions [24].

To relax constraints of the analysis, we looked at a broader scale considering all genes and using PFAM domain annotations. Similar to the above analysis, heterogenic profiles were observed in that corals were clearly distinct from other cnidarians but diverse among themselves. It is especially noteworthy that domain abundances were more consistent between Actiniara (*Aiptasia*) and Corallimorpharia (Afen, Disco), than within Scleractinia (Additional file 1: Figure S14). Nevertheless, corals

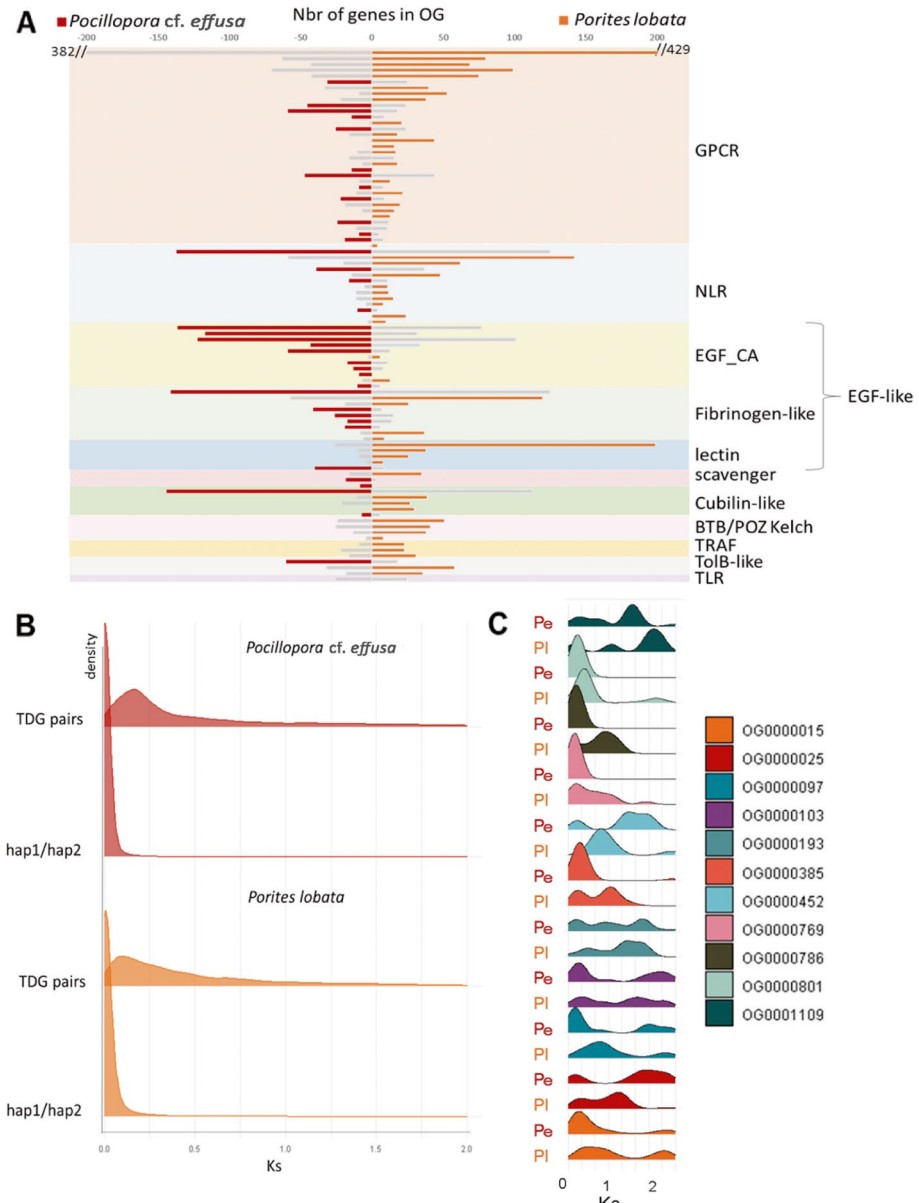

**Fig. 5 A** Comparison of the number of genes in amplified gene families of *Porites lobata* and *Pocillopora* cf. *effusa* genomes. Gene families are grouped by their functional annotation. The largest gene family is indicated by a colored bar, for *P.* cf. *effusa* (red) and *P. lobata* (orange). **B** $K_s$ distribution of *Porites lobata* and *Pocillopora* cf. *effusa* tandemly duplicated gene pairs (TDG) and allelic gene pairs (BRH between haplotype 1 and haplotype 2 annotations "hap1/hap2"). **C** $K_s$ distributions for TDG pairs in *P. lobata* (Pl) and *P.* cf. *effusa* (Pm) for 11 orthogroups (OG) that are amplified in corals and contain NACHT domains

of the genus *Porites* and family Pocilloporidae clustered together. Thus, even at the domain level, corals were diverse with regard to amplified functional families. However, in a broader view, the gene functions of the amplified domains could be assorted majoritively (Additional file 3) to signalling pathways. Glycosyl transferase domains stand out as very enriched in corals. Such domains were shown to be amplified in various coral species [26] and to be associated with NACHT domains (NLR proteins) and TIR domains (TLR proteins) in *Acropora* [53].

### Comparison of gene family amplifications within corals

To date, only two biosynthetic differences between robust and complex have been reported [22, 26] and few biological criteria resolve the two groups. Here, based on the same method as for the coral vs sea anemones comparison, we identified OG that are differentially expanded in robust vs complex species. We obtained a list of 38 OG amplified in robust species and 43 OG amplified in complex species (Additional files 4 and 5). Again, we observe that the differences between gene abundance occur mainly at the genus level rather than transcend to the level of robust or complex lineages (Additional file 1: Figure S15), i.e., that gene expansions are genus- or species-specific. For instance, OG0000628 (G-protein-coupled receptor, rhodopsin-like) is strongly abundant in *Porites lobata* but not in other complex species and OG0000316 (C-type lectin-like) is abundant in *Pocillopora* cf. *effusa* but not in other robust species. Considering all genes and taking PFAM domain annotation into account, analysis of enriched PFAM domains shows that although corals broadly separate into complex and robust clades, coral species within their respective clades exhibit differences with regard to domain abundance. Notably, *Pocilloporidae* corals within the robust clade as well as *Porites* corals within the complex clade cluster together, suggesting that besides the substantial differences between species, conserved patterns that align with phylogeny at various levels are perceptible (Additional file 1: Figure S16 and Additional file 6).

Additionally, we tried to discriminate coral species based on the morphological distinction of massive and branched coral colonies, since it has been described that massive colonies are more tolerant to bleaching than branched colonies [54–56]. We obtained a list of 65 OG amplified in massive species and 20 OG amplified in branched species (Additional file 1: Figure S17, Additional files 7 and 8). Even if the amplified gene families have similar functions, the situation is much more imbalanced compared to the robust/complex comparison. This observation may suggest the functional link between the observed gene amplification (especially disease resistance and immune gene-pools) and the resilience of massive species.

Since the number of genes annotated in all species might not be comparable (especially for TDG in genomes sequenced with short reads), we compared the two genomes that we sequenced with long reads and annotated with the same procedure, *Porites lobata* and *Pocillopora* cf. *effusa*. We show that among the 192 OG amplified in corals, most display higher gene copy numbers in *Porites lobata* than *Pocillopora* cf. *effusa* (respectively 117 and 64: Additional file 2), which is in agreement with the observation that more TDG are detected in *Porites lobata* than *Pocillopora* cf. *effusa*. However, OG containing EGF-like domains, and especially EGF_CA (calcium-binding EGF domain), are more abundant in *Pocillopora* cf. *effusa* than *Porites lobata* (Fig. 5A). This domain has previously been shown to be abundant in the extracellular matrix [57]. But again, the lack of long-read assemblies makes it difficult to determine whether these differences are a general trait in complex and robust corals or massive and branched corals.

### Mechanisms of gene amplification and evolution of amplified genes

We investigated evolutionary rate variation of expanded gene families using CAFE [58]. We reconstructed the number of genes in each OG for internal nodes of the

phylogenetic tree and identified significant increases or decreases in gene copy numbers. It is notable that more events have occurred on branches leading to coral species than to other hexacorallia. These events are occurring on various branches of the phylogenetic tree and each OG has a different history (Additional file 1: Figure S18), which is in accordance with the variety of gene abundance profiles already observed (Fig. 4B). We also detected a large number of gene amplifications that are species-specific. However, the species for which the highest number of gene amplifications is detected by CAFE is *Porites lobata*, which likely reflects the higher number of annotated TDG compared to other *Porites* (Additional file 1: Figure S19). This result highlights the difficulty to conduct such analyses of amplification/reduction with species having heterogeneous gene prediction exhaustivity.

To trace the evolutionary history of these duplicated genes, we calculated the synonymous substitution rates (Ks) between pairs of tandemly duplicated genes in *Porites lobata* and *Pocillopora* cf. *effusa*. Ks are used to represent the divergence time between duplicated copies (lower Ks reflect higher divergence). Although a single peak in the Ks distribution is distinguishable for the two species (Fig. 5B), the degrees of conservation are highly variable within orthogroups (Fig. 5C) or even between tandemly duplicated gene clusters inside one orthogroup (Fig. 6E). When looking at an example of gene family amplified in corals (TIR-domain-containing) (Fig. 6), we observe that some TDG predate the scleractinian/non-scleractinian divergence, and others are specific of robust or complex clades, or of *Porites* (Fig. 6B) or *Pocilloporidae* (Fig. 6C). As expected, for TDG shared by more species (more ancient), Ks are higher (Fig. 6E). Gene family amplification by tandem duplication thus appears to be a dynamic process that has been occurring for a long time and is still at play. We propose that, in corals, the main gene family expansion mechanism is birth-and-death evolution [59]. Birth of new copies occurs by tandem duplications at the genomic level, which is in accordance with the observation that more distant gene pairs on the genome show lower conservation (Additional file 1: Figure S20). This mechanism of duplication at the genomic level is consistent with the existence of duplications of groups of two or three adjacent genes together (Additional file 1: Figure S21). Tandem duplication is followed by divergence of the gene copies, especially in intronic sequences. Indeed, when comparing structures of duplicated gene pairs, the vast majority show conserved exon lengths unlike introns (only 11 *Porites lobata* and 21 *Pocillopora* cf. *effusa* gene pairs show perfectly conserved exon and intron structures), and the sequence conservation of introns is lower than that of exons (Additional file 1: Figure S22 and Additional file 1: Figure S23).

### Transcription profile of amplified genes in environmental samples

Gene duplication plays a key role in the creation of novelty [46] (including new gene functions) but can also, when associated with regulatory mutations, affect gene expression and lead to new expression patterns [60]. These changes in gene expression may underlie much of phenotypic evolution [61]. Therefore, comparing the expression patterns of both old and new duplicate genes can provide information about their functional evolution. Here, we quantified the abundance of *Pocillopora* cf. *effusa* transcripts in 103 available environmental samples coming from 11 different islands of the Tara Pacific expedition [62, 63] (Fig. 7A).

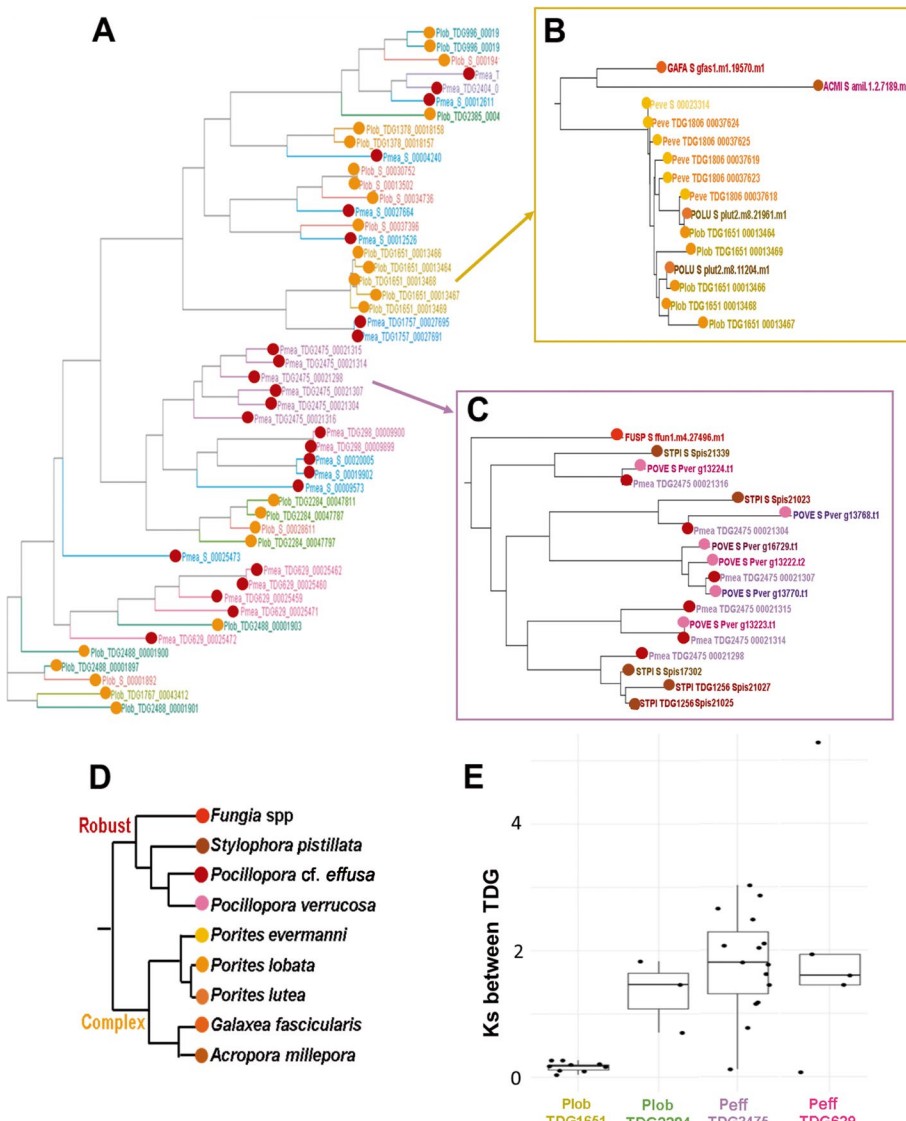

**Fig. 6  A** Approximately maximum-likelihood phylogenetic tree obtained with FastTree after aligning proteins from OG0000106 (TIR-domain-containing orthogroup) in *Porites lobata* (orange dots) and *Pocillopora* cf. *effusa* (red dots). Colors correspond to tandemly repeated gene clusters (singletons are in red). **B,C** Trees obtained for 15 coral species for two individual TDG clusters. Dot colors correspond to species displayed in the species tree in **D**. **E** Distribution of $K_s$ between pairs of genes in TDG clusters, in *P. lobata* and *P.* cf. *effusa*

Of the 32,095 genes of *P.* cf. *effusa*, 94.3% are expressed in at least one environmental sample. This proportion is even higher for TDG (97.4%) underlining the strong support of these particular genes. We then examined the expression of the duplicated genes across the 103 samples and found highly correlated expression patterns in the case of recent duplicated genes (average Pearson 0.6) compared to old gene duplications (average Pearson 0.2, Fig. 7B). This observation highlights the importance of mutations in gene expression changes after gene duplications. The subfunctionalization of expression is described as a rare event [64], and usually recent duplicates are downregulated in accordance with the dosage-sharing hypothesis. Data generated in the frame of the Tara Pacific expedition did not allow us to

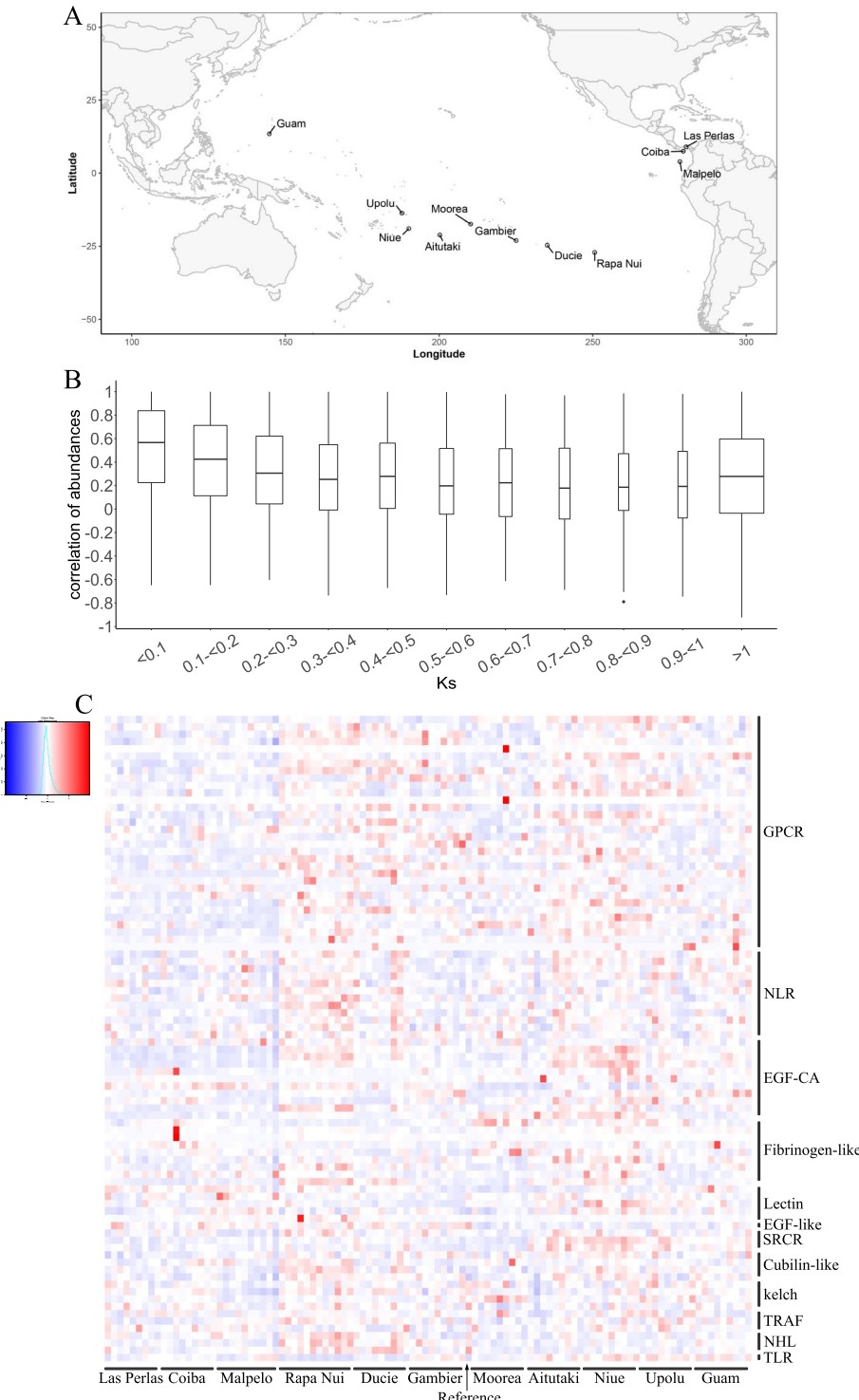

**Fig. 7** **A** Map showing the 11 islands sampled during the Tara Pacific expedition. **B** Pearson correlations of gene expression profiles across the 103 samples for different values of dS between pairs of TDG. **C** Heatmap of expression quantification (*z*-score of mean TPM per OG) of amplified genes in coral genomes across the 103 samples. The column named Reference corresponds to RNA extracted from the same individual as that used for genome sequencing

investigate the expression dosage of these duplicate genes at the colony level, because one cannot exclude the existence of differences in the number of copies of a given gene between the colonies, but we suggest that the high number of duplicated genes in corals makes them interesting models to study these questions. Additionally, we investigated the expression profiles of the genes from the 192 amplified OG in coral genomes. We did not find significant differences between gene function or provenance of transcriptomic samples, except for the island of Rapa Nui. Indeed, amplified genes appear to exhibit increased expression in all the samples of this island (Fig. 7C), which is also the case for non-amplified genes (Additional file 1: Figure S24). Interestingly, the Rapa Nui island exhibits a unique association of coral and symbiont lineages [65] as well as short telomeric DNA associated to an overexpression of several telomere maintenance genes [66].

### Amplification of gene families functionally important for corals

#### *Innate immune system receptors*

Most of the gene families that are amplified in coral genomes correspond to receptors that are likely related to self/non-self recognition and play a role in innate immunity or host-symbiont interactions [67, 68]. Membrane pattern recognition receptors (PRRs) play a central role in immune recognition in cnidarians [68–70]. Toll-like receptors (TLRs) are transmembrane receptors containing TIR (Toll/Interleukin-1 receptor) domain, leucine-rich repeats (LRRs), and a cysteine-rich domain [71, 72]. Lectins, including tachylectins [73], also act as PRRs and are involved in activation of the complement cascade. NOD-like receptors function as cytosolic receptors and contain NACHT/NB-ARC domains [53]. These gene families have previously been shown to be amplified in corals compared to other cnidaria [71, 25, 22]. We identified TIR, NACHT and lectin containing gene families based on their domain composition and counted the number of corresponding genes in each coral species (Additional file 9). As seen for the 192 OG amplified in corals (Fig. 4B), each species/genus displays specific amplified innate immunity gene families (Additional file 1: Figure S25). This observation is in agreement with earlier studies that have shown that coral species diverge on which innate immunity-related genes are expanded, and have also suggested that innate immune pathways might play diverse adaptive roles [24, 25]. Strikingly, *Porites* species display the highest number of gene copies when cumulating the three innate immune system receptor categories studied here (Fig. 8A). It is tempting to speculate that this huge repertoire of innate immune system genes could play a role in the notably long lifespan and high resilience of *Porites* species. Interestingly, we also identified a high number of GPCR-like (G-protein-coupled receptor) proteins among coral-amplified gene families: the function of this very abundant transmembrane receptor family in corals will require further investigation, but it is likely to be involved in innate immunity and/or host-symbiont interactions.

We studied the domain composition of TIR-containing and NACHT/NB-ARC containing proteins in 5 coral species, and confirmed the high number of domain combinations already observed in *Acropora digitifera* [53]. Interestingly, our high-quality assembly of *Porites lobata* allowed the annotation of a high number of IL-1R-like proteins (containing TIR domains associated with Immunoglobulin domains). We also identified SARM-like proteins shared between all corals and confirmed the expansion of TIR-only proteins in corals [71] (Fig. 8B).

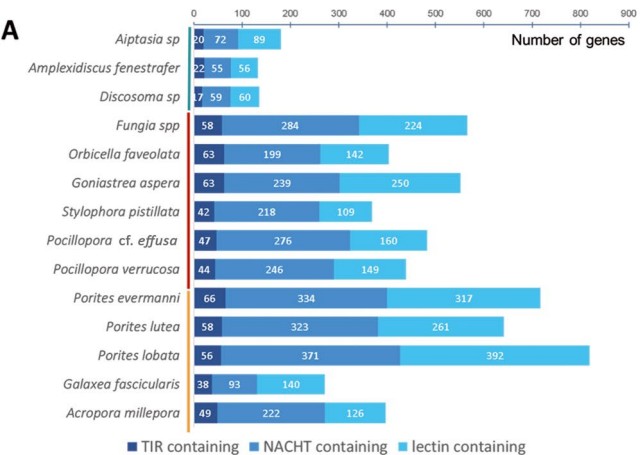

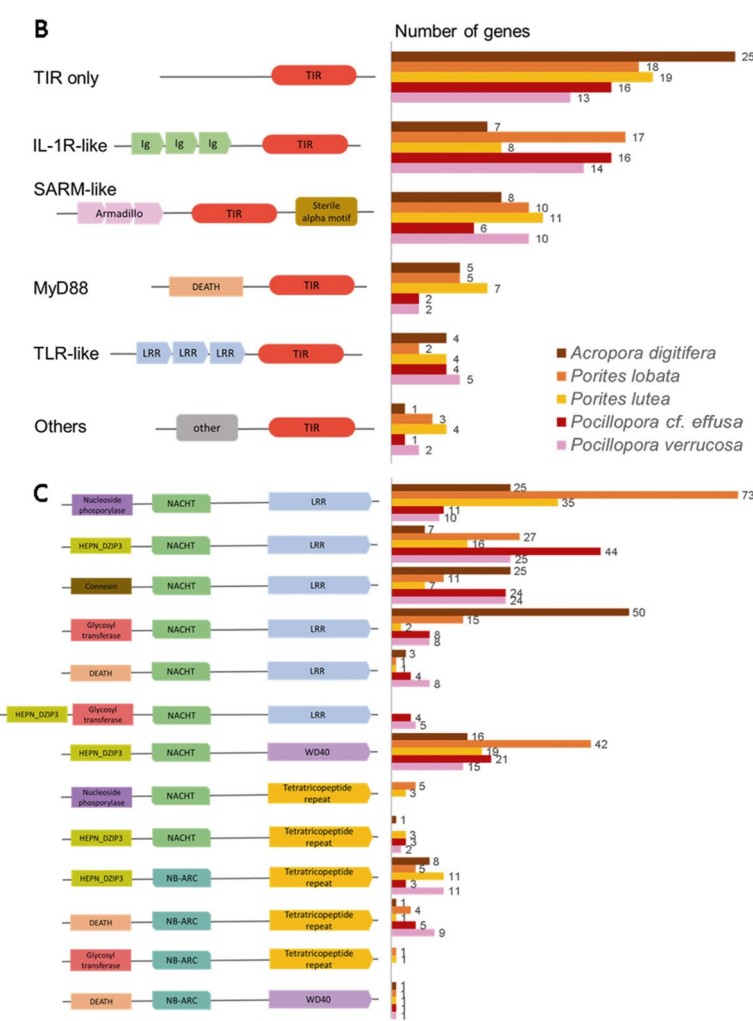

**Fig. 8** **A**. Cumulative bar plot representing the number of genes in innate immune receptor OG identified from domain annotation of 14 cnidarian gene sets, for three innate immune receptor categories. **B,C** Domain composition of TIR-containing (**B**) and NACHT/NB-ARC (**C**) containing proteins in 5 coral species. Left panels: schematic view of domain composition. Right panels: number of genes in each species

*Porites lobata* contains a much lower proportion of truncated (lacking N- and/or C-terminal regions) NACHT/NB-ARC containing genes than *Porites lutea* or *Acropora digitifera*, where a high number of NACHT/NB-ARC-only proteins were described [53]: we hypothesize that a substantial fraction of these proteins corresponds to truncated annotations (Additional file 1: Figure S26). As already observed for NOD-like receptors in *Acropora digitifera*, the effector and C-terminal repeated domains found are distinct between NACHT-containing (mostly LRR and WD40 repeats) and NB-ARC-containing (mostly tetratricopeptide repeats) proteins (Additional file 1: Figure S27). Distinct domain combination abundances are observed in the different coral species, and some are species-specific (Fig. 8C). A high number of N-terminal RNA-binding "DZIP3-like HEPN" domains are observed. HEPN domains are found in a variety of defense and stress response systems across the tree of life and could play a role in antiviral, antitransposon, apoptotic systems, or RNA-level response to unfolded proteins [74].

### Calcification-related genes

Corals build the structural foundation of coral reefs through calcification. Genes and their associated functions involved in this process are already well characterised [75]. Here, we examined seven families of candidate genes encoding a set of proteins involved in coral calcification [57, 76]. Among these families, proteins can be divided into 2 categories on the basis of their role and localization in calcification: ion membrane transporters/enzymes and skeletal organic matrix proteins (Additional file 1: Figure S28). The first category comprises ammonium transporters Amt1[77] (Additional file 1: Figure S29 and Additional file 1: Figure S30); the Bicarbonate Anion Transporter SLC4 gamma [76, 78] (Additional file 1: Figure S31); plasma-membrane calcium ATPases [57, 78] (PMCAs); and carbonic anhydrases [75, 79, 80] (CAs). CAs also fall within the category of organic matrix proteins since one isoform is found in coral skeletons [81]. Organic matrix proteins comprise coral acid-rich proteins [82, 83] (CARPs) and neurexin [84].

We compared these seven families in the genomes of 12 scleractinian (6 robust and 6 complex) and 3 non-scleractinian species (two Corallimorphs and one Actiniaria) and found different evolutionary histories (Additional file 1: Table S10). As observed previously [57], calcification-related genes are often clustered in tandem in coral genomes (Additional file 1: Figure S30 and Additional file 1: Figure S31). Scleractinian divide into robust and complex clades, which diverged about 245 Mya and show different skeletal properties [57]. They diverged from Actiniaria approximately 506 Mya and from Corallimorpharia about 308 Mya at the time they acquired the ability to calcify [57]. First, our results show that a set of proteins (PMCAs, neurexin) are present in the proteomes of all *Pocillopora* and *Porites* as well as other Hexacorallia with no significant difference in number. Second, they possess the SCL4γ, which is scleractinian specific and is a tandem duplication of the SLC4β gene. Finally, *Pocillopora* possesses a higher number of orthologs of CARPs, Amt1 and CAs than *Porites* (Additional file 1: Table S7 and Additional file 1: Figure S28). Our results clearly confirm that gene amplification for this set of calcification-related gene families differ between robust and complex but show co-option of genes and neofunctionalization for calcification that occurred during evolutionary history of species.

## Discussion

The introduction of long-read sequencing is a game changer for obtaining complete reference genome sequences. Short-read sequencing has long been considered sufficient to access the gene catalog of a given species. Herein we have shown, using two coral genomes from two different groups, that long-read technologies are essential to obtain an accurate view of the gene content and the functional landscape of genomes. Notably, long reads validate the tandem organization of duplicated genes identified by means of our dedicated annotation method. Thereby, we highlight pervasive tandem gene duplications in coral genomes. This peculiarity was previously underestimated due to the difficulty of assembling repetitive regions with short reads and the greater fragmentation of the resulting assemblies which did not allow the detection of large blocks of TDG. Moreover, the high level of heterozygosity in coral genomes complicates the detection of duplicated genes. Indeed, the remaining allelic duplications isolated on small contigs can be confused with true duplicated genes.

Based on these complete gene catalogs of corals, we observe large (and ancient) arrays of TDG that remained clustered on the genome over a long time. This situation is quite unusual compared to what is observed in plant genomes for instance, where only recent TDG are clustered together and ancient copies have been translocated to other chromosomes [49, 85]. Translocation of duplicated genes has been proposed to be a means to escape concerted evolution, that is homogenization of gene copies through gene conversion [86]. In corals, however, gene conversion does not appear to be a common mechanism, since very few gene pairs happen to show strongly conserved exon/intron structures, and even genes that are tightly linked on the genome accumulate mutations (mostly in introns). Accordingly, in the last decades, the growing availability of genomic data revealed that most multigene families display high levels of intraspecific diversity, which is not consistent with a homogenizing mechanism [87]. The currently accepted model of multigene family evolution is the birth-and-death evolution model, first proposed by Nei and colleagues [59, 88]. This model is in accordance with what is observed in coral gene families, where tandem duplication appears to be a dynamic process that has been taking place for a long time and is still ongoing. The fact that tandemly repeated genes have remained clustered together could be related to the high synteny conservation between coral species. This observation suggests that gene translocation (leading to loss of synteny) is not needed to allow genes to diverge and evolve new functions, since gene conversion is very rare.

The high number of TDG in corals and their maintenance in arrays suggest that these species are excellent models to study tandem duplications in genomes, although this complicates the generation of contiguous genome assemblies. Sequencing and annotating more coral genomes at chromosome scale and a high-quality level, especially outside *Porites* and *Pocillipora* genera, is important. This will first give a higher resolution of the scleractinia order, then will be necessary to decipher how and where genomic duplications occur, and the evolution of these gene duplications. Additional transcriptomic resources will also be needed to make it possible to investigate the fate of duplicated copies, in particular the impact of duplicated genes on the expression dosage. Global gene expression analysis of *Pocillopora* has revealed a high transcriptomic plasticity dependent on both the genetic lineage and the environment [65]. It is tempting to draw

a parallel with our observation of divergent expression profiles between duplicated genes and suggest that these gene expression patterns may play a role in the acclimatization capacities of *Pocillopora* to the environment.

Our comparative analysis based on available coral genomes reveals a high number of amplified gene families compared to sea anemones. We show that these gene families are functionally related to signal reception and transduction, especially innate immunity. Several studies have shown that some of these families play a key role in maintaining the symbiosis of corals with their associated Symbiodiniaceae [52, 89]. We also noted that these families are mainly amplified through tandem duplications, and their retention in the genome through evolution underlines their functional importance in corals. Gene duplication has already been described as playing an important role in phenotypic evolution: in particular, a link with long lifespan has already been reported in other organisms, such as trees and fishes. Indeed, a convergent expansion of disease-resistance gene families across several tree species suggests that the immune system contributes to the survival of long-lived plants [49]. Similarly, gene expansion of immunoregulatory genes in rockfishes may have facilitated adaptations to extreme life span [90]. We hypothesize that these amplified gene families in corals, related to innate immunity and disease resistance, may have contributed to the resilience and long lifespan of these sessile organisms. Our analyses also revealed a short sequence of 188-bp tandemly repeated in several intergenic regions that is uniquely found in *Porites* genomes. This satellite-like sequence is intriguing because it contains a palindromic telomeric sequence. Since telomeres are key ageing hallmarks in numerous organisms [91], it is tempting to speculate that it is involved in the stress resistance and extreme longevity features shared by *Porites* species [39].

Precise identification of the timing of duplications for each individual gene is hampered by the fact that current coral genome assemblies and annotations have missed some duplicated genes. However, we found ancient duplicated genes that are shared by all corals, but also recently duplicated genes that are species- or genus-specific. Even though functions of amplified gene families are common across coral species, the individual genes that are amplified can be different. This striking pattern is a neat example of convergent evolution and can be seen as an important evolutionary advantage. Differences between coral species, namely which genes have been amplified as well as which new expression patterns have emerged within duplicated genes, could provide them different abilities to overcome environmental changes.

## Conclusions

We generated two highly contiguous genome sequences with complete gene catalogs that allowed us to highlight pervasive tandem gene duplications in stony corals. These duplicated genes originate from multiple and distinct duplication events, through a birth and death evolution process, and contribute to the amplification of gene families functionally related to signal reception and transduction, especially innate immunity. Moreover, each coral lineage appears to have developed a specific repertoire of innate immunity genes through convergent evolution. This variety of amplified genes could give corals a wider range of response to environmental changes. This plasticity (at gene content and expression pattern levels) may therefore represent an evolutionary opportunity

for coral species, and involved genes are potential targets for assisted evolution of more resilient corals [52, 92, 93].

## Methods

### Coral material and species names

*Porites evermanni*, *Porites lobata*, and *Pocillopora* cf. *effusa* colonies were collected in French Polynesia by the CRIOBE at Moorea. The assembled genome of *Pocillopora* was assigned to SVD1 lineage based on genome-wide SNPs (Voolstra et al. in revision). The SVD1 lineage corresponds to GSH01 [94] and carries the mtORF marker type 2/11. Therefore, this genome is *Pocillopora* cf. *effusa*. Fifty grams of each coral colony was flash frozen using liquid nitrogen and stored at − 80 ℃ until further use. In addition, a small fragment of *Porites lobata* was placed in 2 ml of Lysing Matrix A beads (MP Biomedicals, Santa Ana, CA, USA) in presence of 1.5 ml of DNA/RNA shield (Zymo Research, Irvine, CA, USA) and stored at − 20 ℃ until RNA purification.

### DNA extraction

DNA extractions from flash frozen tissues were based on a nuclei isolation approach to minimize contamination with Symbiodiniaceae DNA. Briefly, cells were harvested using a Waterpik in 50 ml of 0.2 MEDTA solution refrigerated at 4 ℃. Extracts were passed sequentially through a 100-μm then a 40-μm cell strainer (Falcon) to eliminate most of the Symbiodiniaceae. Then extracts were centrifuged at 2000 *g* for 10 min at 4 ℃ and the supernatants were discarded. Two different protocols of DNA purification were used as follows. For *Porites evermanni* and *Pocillopora* cf. *effusa*, the resulting pellets were homogenized in lysis buffer (G2) of the Qiagen Genomic DNA isolation kit (Qiagen, Hilden, Germany). The DNA were then purified following the manufacturer's instructions using genomic tip 100/G. For *Porites lobata*, the pellet was homogenized in CTAB lysis buffer followed by a Chloroform/isoamyl alcohol purification.

### RNA extraction

Total RNA of *Pocillopora* cf. *effusa* was extracted from flash frozen tissue. For *Porites lobata*, the fragment placed in 2 ml of Lysing Matrix A beads (MP Biomedicals, Santa Ana, CA, USA) in presence of 1.5 ml of DNA/RNA shield (Zymo Research, Irvine, CA, USA) was thawed and disrupted by the simultaneous multidirectional striking using a high-speed homogenizer FastPrep-24 5G Instrument (MP Biomedicals, Santa Ana, CA, USA) (following conditions: speed 6.0 m/s, time 30 s, pause time 60 s, cycles 3). Total RNA was then purified from one aliquot of 500 μl of homogenized suspension, following the instruction of the commercial Quick-DNA/RNA Kit (Zymo Research, Irvine, CA, USA).

Approximately 5 μg of total purified RNA was then treated with the Turbo DNA-free kit (Thermo Fisher Scientific, Waltham, MA, USA), according to the manufacturer's protocol. Quantity was assessed on a Qubit 2.0 Fluorometer using a Qubit RNA HS Assay kit, and the quality was checked by capillary electrophoresis on an Agilent Bioanalyzer using the RNA 6000 Pico LabChip kit (Agilent Technologies, Santa Clara, CA, USA). Purified RNA was stored at − 80 ℃ until further use.

### Illumina library preparation and sequencing

Genome sequencing of *Porites evermanni* was performed using Illumina reads from both paired-end and mate-pair (MP) libraries of different insert sizes. The paired-end library was prepared using the NEBNext DNA Modules Products (New England Biolabs, MA, USA) with a "on beads" protocol developed at the Genoscope, as previously described [95] in Alberti et al. The library was quantified by qPCR (MxPro, Agilent Technologies) using the KAPA Library Quantification Kit for Illumina Libraries (Roche), and its profile was assessed using a DNA High Sensitivity LabChip kit on an Agilent 2100 Bioanalyzer (Agilent Technologies, Santa Clara, CA, USA). The library was paired-end sequenced on the Illumina HiSeq2500 (Illumina, USA) sequencing platform ($2 \times 251$ bp). The MP libraries were prepared using the Nextera Mate Pair Sample Preparation Kit (Illumina, San Diego, CA). Briefly, genomic DNA (4 µg) was simultaneously enzymatically fragmented and tagged with a biotinylated adaptor. Tagmented fragments were size-selected (3–5; 5–8; 8–11 and 11–15 kb) through regular gel electrophoresis, and circularized overnight with a ligase. Linear, non-circularized fragments were digested and circularized DNA was fragmented to 300–1000-bp size range using Covaris E220. Biotinylated DNA was immobilized on streptavidin beads, end-repaired, and 3′-adenylated. Subsequently, Illumina adapters were ligated. DNA fragments were PCR-amplified using Illumina adapter-specific primers and then purified. Finally, libraries were quantified by qPCR and library profiles were evaluated using an Agilent 2100 Bioanalyzer. Each library was sequenced using 151 base-length read chemistry on a paired-end flow cell on the Illumina HiSeq4000 sequencing platform (Illumina, USA).

The genomes of *Porites lobata* and *Pocillopora* cf. *effusa* were sequenced using a combined approach of short and long reads. The short reads were obtained by preparing Illumina PCR-free libraries using the Kapa Hyper Prep Kit (Roche). Briefly, DNA (1.5 µg) was sonicated to a 100- to 1500-bp size range using a Covaris E220 sonicator (Covaris, Woburn, MA, USA). Fragments were end-repaired, 3′-adenylated, and Illumina adapters were ligated according to the manufacturer's instructions. Ligation products were purified with AMPure XP beads (Beckman Coulter Genomics, Danvers, MA, USA) and quantified by qPCR. Library profiles were assessed using an Agilent High Sensitivity DNA kit on the Agilent 2100 Bioanalyzer. Libraries were paired-end sequenced on an Illumina HiSeq2500 instrument (Illumina, San Diego, CA, USA) using 251 base-length read chemistry.

Illumina RNA-seq libraries were prepared for both *Porites lobata* and *Pocillopora* cf. *effusa* using the TruSeq Stranded mRNA kit (Illumina, San Diego, CA, USA), according to the manufacturer's protocol starting with 500 ng total RNA. Briefly, poly(A) + RNA were selected with oligo(dT) beads, chemically fragmented and converted into single-stranded cDNA using random hexamer priming. Then, the second strand was generated to create double-stranded cDNA. The resulting cDNAs were subjected to A-tailing, adapter ligation, and PCR-amplification. Ready-to-sequence Illumina libraries were then quantified by qPCR and library profiles evaluated with an Agilent 2100 Bioanalyzer. Each library was sequenced using 151-bp paired-end reads chemistry on a HiSeq4000 Illumina sequencer. Short Illumina reads were bioinformatically post-processed sensu Alberti et al. [95] to filter out low-quality data. First, low-quality nucleotides ($Q < 20$)

were discarded from both read ends. Then remaining Illumina sequencing adapters and primer sequences were removed and only reads ≥ 30 nucleotides were retained. These filtering steps were done using in-house-designed software based on the FastX package [96]. Finally, read pairs mapping to the phage phiX genome were identified and discarded using SOAP aligner [97] (default parameters) and the Enterobacteria phage PhiX174 reference sequence (GenBank: NC_001422.1).

### MinION and PromethION library preparation and sequencing

Genomic DNA fragments of *Pocillopora* cf. *effusa* ranging from 20 to 80 kb were first selected using the Blue Pippin (Sage Science, Beverly, MA, USA) then repaired and 3′-adenylated with the NEBNext FFPE DNA Repair Mix and the NEBNext® Ultra™ II End Repair/dA-Tailing Module (New England Biolabs, Ipswich, MA, USA). Sequencing adapters provided by Oxford Nanopore Technologies (Oxford Nanopore Technologies Ltd, Oxford, UK) were then ligated using the NEBNext Quick Ligation Module (NEB). After purification with AMPure XP beads, the library was mixed with the Sequencing Buffer (ONT) and the Loading Beads (ONT). For *Pocillopora* cf. *effusa*, a first library was prepared using the Oxford Nanopore SQK-LSK108 kit and loaded onto a R9.5 MinION Mk1b flow cell. Three other libraries were prepared using the same kit and following the same protocol, but without the size selection. They were loaded onto R9.4.1 MinION Mk1b flow cells. An additional library was prepared using the Oxford Nanopore SQK-LSK109 kit and loaded onto a R9.4.1 PromethION flow cell. Reads were basecalled using Albacore version 2. For *Porites lobata*, eight libraries were prepared using the Oxford Nanopore SQK-LSK109 kit. Four libraries were loaded onto R9.4.1 MinION Mk1b flow cells and the other four onto PromethION R9.4.1 flow cells. Reads were basecalled using Guppy version 2. In both cases, the resulting raw nanopore long reads were directly used for the genome assembly.

### Short-read-based genome assembly

The genome of *Porites evermanni* was assembled with megahit [98] using the filtered high-quality Illumina technology paired-end reads from shotgun libraries (Additional file 1: Table S1). The resulting assembly was scaffolded using mate-pair libraries and SSpace [99] and gap-filled with gapcloser [100] and the paired-end reads. This process generated an assembly of 604 Mb composed of 8186 scaffolds with an N50 of 171 Kb (Table 1).

### Long-read-based genome assemblies

To generate long-read-based genome assemblies, we generated three samples of reads: (i) all reads, (ii) 30X coverage of the longest reads, and (iii) 30X coverage of the filtlong [101] highest-score reads that were used as input data for four different assemblers, Smartdenovo [102], Redbean [103], Flye [104], and Ra (Additional file 1: Table S2). Smartdenovo was launched with -k 17 and -c 1 to generate a consensus sequence. Redbean was launched with "-xont -X5000 -g450m" and Flye with "-g 450 m." The resulting assemblies were evaluated based on the cumulative size and contiguity, with the Smartdenovo and all read combination producing the best assembly. This assembly was polished three times using Racon [105] with Nanopore reads, and two times with Hapo-G [106] and Illumina PCR-free reads.

### Reconstruction of allelic relationships and haploid assembly

The cumulative size of both assemblies was higher than expected due to the high heterozygosity rate (Additional file 1: Figure S1), but also the presence of several organisms that can bias kmer distributions. Here, we found a few contigs that correspond to the mitochondrial genome of symbiotic algae (section " Contamination removal"). As indicated by BUSCO [107] and KAT [108], we observed the two alleles for many genes and a significant proportion of homozygous kmers were present twice in the assembly. We used Haplomerger2 with default parameters and generated a haploid version of the two assemblies. Haplomerger2 detected allelic duplications through all-against-all alignments and chose for each alignment the longest genomic regions (parameter −selectLongHaplotype), which may generate haplotype switches but ensure to maximize the gene content. We obtained two haplotypes for each genome: a reference version composed of the longer haplotype (when two haplotypes are available for a genomic locus) and a second version, named alternative, with the corresponding other allele of each genomic locus. Consistently keeping the longest allele in the reference haplotype explains the larger size of the reference assembly. As an example for *Porites lobata*, assemblies had a cumulative size of 642 and 588 Mb for the reference and the alternative assembly version, respectively (Additional file 1: Table S3). At the end of the process, the *P. lobata* and *P.* cf. *effusa* haploid assemblies have a cumulative size of 650 and 350 Mb respectively, closer to the expected ones. BUSCO and KAT analysis showed a reduction of the allelic duplications (Additional file 1: Table S5, Additional file 1: Figure S2 and Additional file 1: Figure S3). Final assemblies were polished one last time with Hapo-G [106] and Illumina short reads to ensure that no allelic regions present twice in the diploid assembly have remained unpolished.

Additionally, we compared our haploid genome assemblies with the one obtained using the Purge Dups tool [51]. We sampled 50X of the longest Nanopore reads and launched Purge Dups on the raw Nanopore assemblies with default parameters (Additional file 1: Table S3). For *Porites lobata*, we obtained an assembly of 588 Mb composed of 1057 contigs. By comparison, the *Pocillopora* cf. *effusa* raw assembly was not altered by Purge Dups while the coverage thresholds were consistent. This may be due to the lower proportion of haplotypic duplications (Additional file 1: Figure S4). BUSCO scores and kmer distributions (Additional file 1: Table S5 and Additional file 1: Figure S4) were very similar for both Haplomerger2 and Purge Dups assemblies for *Porites lobata* which is a confirmation of the great work performed by haplomerger2.

### Contamination removal

As we used a DNA extraction method based on a nuclei isolation approach, we minimized contamination with DNA from other organisms (most notably, Symbiodiniaceae). We predicted coding fragments on both assemblies using metagene and aligned their corresponding protein sequences against the nr database. Contigs were classified based on their hits, and contigs with more than 50% of genes having a best hit to bacteria, archaea, or viruses were classified as non-eukaryotic and filtered out. In addition, contigs taxonomically assigned to Symbiodiniaceae were also filtered out. In the *Porites lobata* assembly, we filtered 38 contigs with an average size of 25 kb and totaling 961 kb,

while in the *Pocillopora* cf. *effusa* assembly, only two contigs of 25 kb and 30 Kb were filtered out.

### Transcriptome assembly

First, ribosomal RNA-like reads were detected using SortMeRNA (Kopylova et al., 2012) and filtered out. Illumina RNA-Seq short reads from *Porites lobata* and *Pocillopora* cf. *effusa* were assembled using Velvet [109] 1.2.07 and Oases [110] 0.2.08, using a kmer size of 89 and 81 bp respectively. Reads were mapped back to the contigs with BWA-mem, and only consistent paired-end reads were kept. Uncovered regions were detected and used to identify chimeric contigs. In addition, open reading frames (ORF) and domains were searched using respectively TransDecoder and CDDsearch. Contigs were broken in uncovered regions outside ORF and domains. At the end, the read strand information was used to correctly orient RNA-Seq contigs.

### Repeat detection

Libraries of genomic repeats were first detected using RepeatModeler (v.2.0.1, default parameters) on both genomes. Then, these libraries were annotated with RepeatMasker [111] (v.4.1.0, default parameters) and RepBase (from RepeatMasker v4.0.5). Finally, the *P. lobata* and *P.* cf. *effusa* genomes were masked using their respective libraries. The numbers of bases were counted according to the classification of repeat overlapping each base. In case of overlapping repeated fragments, the longest annotated one was selected.

### Gene prediction

Gene prediction was done using proteins from 18 Cnidarian species, *Acropora digitifera*, *Acropora millepora*, *Aiptasia*, *Aurelia aurita* from Atlantic, *Aurelia aurita* from Pacific, *Clytia hemisphaerica*, *Fungia* sp., *Galaxea fascicularis*, *Goniastrea aspera*, *Hydra vulgaris*, *Montipora capitata*, *Morbakka virulenta*, *Nematostella vectensis*, *Orbicella faveolata*, *Pocillopora damicornis*, *Porites lutea*, *Porites rus,* and *Stylophora pistillata* (Additional file 1: Table S5).

The proteomes were aligned against *Porites lobata* and *Pocillopora* cf. *effusa* genome assemblies in two steps. Firstly, BLAT [112] (default parameters) was used to quickly localize corresponding putative genes of the proteins on the genome. The best match and matches with a score $\geq$ 90% of the best match score were retained. Secondly, the alignments were refined using Genewise [113] (default parameters), which is more precise for intron/exon boundary detection. Alignments were kept if more than 50% of the length of the protein was aligned to the genome. In order to reduce mapping noise, for each proteome mapping alignments without introns are removed if they represent more than 40% of the number of alignments. Moreover, alignments containing at least one unique intron (i.e. intron detected using only one proteome alignements) are removed if they cover at least 10 exons detected in all alignments using all proteomes.

To allow the detection of expressed and/or specific genes, we also aligned the assembled transcriptomes of each species on their respective genome assembly using BLAT [112] (default parameters). For each transcript, the best match was selected based on the alignment score. Finally, alignments were recomputed in the previously identified genomic regions by Est2Genome [114] in order to define precisely intron boundaries.

Alignments were kept if more than 80% of the length of the transcript was aligned to the genome with a minimal identity percent of 95%.

To proceed to the gene prediction for both species, we integrated the protein homologies and transcript mapping using a combiner called Gmove [115]. This tool can find CDSs based on genome located evidence without any calibration step. Briefly, putative exons and introns, extracted from the alignments, were used to build a simplified graph by removing redundancies. Then, Gmove extracted all paths from the graph and searched for open reading frames (ORFs) consistent with the protein evidence. Single-exon genes with a CDS length smaller or equal to 100 amino acids were filtered out. From the remaining genes, only genes with homologies against more than one species (Diamond [116] v0.9.24, blastp, e-value $\leq 10^{-10}$) or spliced genes with a ratio CDS length / UTR length greater or equal to 0.75 were kept. Then, putative transposable elements (TEs) set were removed from the predicted gene using three different approaches: (i) genes that contain a TE domain from Pfam [117]; (ii) transposon-like genes detected using TransposonPSI (http://transposonpsi.sourceforge.net/, default parameters); (iii) and genes overlapping repetitive elements detected using RepeatMasker [111] and RepeatModeler [118] (v2.0.1, default parameters, repetitive sequence detected by RepeatModeler were annotated using RepeatMasker) on the genome assembly. Also, InterProScan [119] (v5.41–78.0, default parameters) was used to detect conserved protein domains in predicted genes. So, predicted genes without conserved domain covered by at least 90% of their cumulative exonic length, or matching TransposonPSI criteria or selected Pfam domains, were removed from the gene set.

Completeness of the gene catalogs was assessed using BUSCO [107] version 4.0.2 (eukaryota dataset odb10 and default parameters).

The genes of *Pocillopora* cf. *effusa* were previously named using the prefix *Pmea* because we first thought the sampled species was *Pocillopora meandrina*, but recently discovered it was *Pocillopora effusa* instead.

### Gene prediction of alternative haplotypes

Gene prediction of the alternative haplotypes was done using proteins annotated on the reference haplotypes. Proteins were aligned as described previously, using BLAT and Genewise aligners with the same parameters. These alignments were integrated using Gmove as described previously. Additional file 1: Table S4 reports gene catalog statistics for reference and alternative haplotypes. Alignments between genes from the reference and alternative assemblies were computed using DIAMOND and only genes with best reciprocal hits were considered as allelic copies.

### Adaptation of gene prediction workflow for tandemly duplicated genes

The presence of highly conserved genes in the same genomic regions can hinder the gene prediction, mostly if based on the alignments of conserved proteins. Indeed, during the spliced alignment step, individual exons of a given protein sequence can be distributed over several genes. Therefore, in these specific genomic regions, alignments of proteins or RNA-Seq data generally span several genes (with larger introns), which lead to the prediction of chimeric genes and the underestimation of the gene number (Additional file 1: Figure S10). To avoid these gene fusions, we added a step in our workflow. Namely, large introns of spliced alignments obtained with BLAT were post-processed.

For each intron having a size greater than 5 kb, the corresponding alignment was splitted in two inside the intron, and the query sequence was realigned on the two new genomic regions. If the sum of the two alignment scores was greater than the score of the previous alignment, then the two new alignments were kept in place of the alignment that contained the large intron. This process was recursively applied until the sum of the two alignment scores did not satisfy the previous condition. At the end, alignments were refined (with dedicated alignment tools) as described in the Gene Prediction section.

### Telomeric sequences

The interstitial telomeric sequences (ITS) were specifically searched as follows. First, the motif (TTAGGG)4 was searched in the coral genomes using blastn92. The blast result was filtered to keep hits with an identity percentage above 75%, a minimum coverage of 75% and two mismatches maximum were allowed. Distant hits of less than 400 bp were gathered to form a single ITS. The 188-bp satellite sequence was searched in the different coral genomes and in the NT database using Blastn. All the hits had an e-value $< 1e-13$ and an identity percentage $> 80\%$. Then, the matching sequences were used to build a HMM profile, using hmmbuild from HMMER suite [120] (v3.3).

### Detection of tandemly duplicated genes

Protein sets of *Porites lobata*, *Pocillopora* cf. *effusa*, *Porites evermanni*, *Acropora millepora*, *Acropora digitifera*, *Montipora capitata*, *Galaxea fascicularis*, *Porites lutea*, *Pocillopora verrucosa*, *Pocillopora damicornis*, *Stylophora pistillata*, *Goniastrea aspera*, and *Orbicella faveolata* (see references in " Orthogroups and orthologous genes" section) were aligned against themselves using Diamond [121] (v0.9.24). Only matches with an e-value $\leq 10^{-20}$ and 80% of the smallest protein aligned were kept. Two genes were considered as tandemly duplicated if they were co-localized on the same genomic contig and not distant from more than 10 genes to each other. Then, all tandemly duplicated genes were clustered using a single linkage clustering approach.

### Validation of tandemly duplicated genes

We validated the structure of the clusters of TDG, by comparing their overlap with Nanopore long reads. Considering a cluster with three tandemly duplicated genes A, B, and C, we first analyzed the two pairs of adjacent genes A/B and B/C. If at least one Nanopore read completely overlaps genes A and B, we classify the pair A/B as validated. Secondly, we analyzed the whole cluster, in our example, the cluster is validated if at least one Nanopore read overlaps the three genes A, B, and C (Additional file 1: Figure S6). Nanopore reads were mapped using minimap2 and following parameters "-t 36 –sam-hit-only -a -x map-ont" and secondary alignments were filtered using "-F 2308" from samtools. Overlaps between reads and gene positions were computed using bedtool intersect (Additional file 1: Table S9, Additional file 1: Figure S7 and Additional file 1: Figure S8).

### Functional assignment of predicted genes

The derived proteins of *Porites lobata* and *Pocillopora* cf. *effusa* predicted genes were functionally assigned by aligning them against nr from the BLAST Databases distributed

by NCBI (version 25/10/2019) using Diamond [121] (v0.9.24, e-value $\leq 10^{-5}$). Then, for each predicted protein, the best match based on bitscore against RefSeq proteins is selected. If there is no match against RefSeq proteins, then the best match is kept from other matches.

### Orthogroups and orthologous genes

First, we selected the proteins of 25 Cnidarian species: *Acropora millepora, Acropora digitifera, Aiptasia sp., Amplexidiscus fenestrafer, Aurelia aurita* from Pacific*, Aurelia aurita from Atlantic, Clytia hemisphaerica, Dendronephthya gigantea, Discosoma sp., Fungia* sp.*, Galaxea fascicularis, Goniastrea aspera, Hydra vulgaris, Montipora capitata, Morbakka virulenta, Nematostella vectensis, Orbicella faveolata, Pocillopora damicornis, Pocillopora* cf. *effusa, Pocillopora verrucosa, Porites evermanni, Porites lobata, Porites lutea, Porites rus*, and *Stylophora pistillata* (Additional file 1: Table S5)*.* Based on quality metrics, we excluded *Pocillopora acuta* because its number of annotated genes was higher (Fig. 1D) than expected based on comparison to other corals and only a small proportion contained domains (Fig. 1E). The proteomes were aligned against each other using DIAMOND [121] (v0.9.24, e-value $\leq 10^{-10}$, -k 0). Matches were kept only if 50% of the smallest protein length of each pair is aligned. Then, orthogroups (OG) and orthologous genes were built with OrthoFinder [122] (v2.3.11, default parameters). Additionally, OrthoFinder built gene trees for each OG and used them to reconstruct a rooted species tree, that is in agreement with the currently accepted phylogeny of cnidarians (Figs. 1A and 4B). At this stage, we noticed that *Acropora digitifera* and *Montipora capitata* datasets were of lower quality and decided to exclude them from subsequent analyses.

For each orthogroup, we listed the 5 most abundant domains detected with InterProScan on proteins from 25 Cnidarian species (Additional file 10), the 5 most abundant BLASTP hits on nrprot for *Pocillopora* cf. *effusa* proteins (Additional file 11) and the 5 most abundant BLASTP hits for *Porites lobata* proteins (Additional file 12). We inspected these lists manually to assign the most likely function for the 192 orthogroups amplified in corals (Additional file 2).

### Coral synteny

For each genome comparison, OG were used to build syntenic clusters using orthodotter (https://www.genoscope.cns.fr/orthodotter/). Only genomic contigs containing at least 5 genes with orthologs are selected. Co-localized orthologous genes less than 15 other orthologous genes apart are considered as belonging to the same syntenic cluster. A cluster has to be formed by at least 5 syntenic genes. Dotplots for the analysis of synteny in coral genomes were built using orthodotter, and circular views of syntenic regions were generated using Circos [123].

### OG consensus construction

For each of the 27,826 orthogroups (OG) containing at least one coral species, the following steps were applied. Coral proteic sequences were extracted and aligned using Muscle [124] (version 3.8.1551, default parameters). Then, the multiple alignment was filtered using OD-Seq [125] (version 1.0) to remove outlier sequences, with parameter –score—i.e., threshold for outliers in numbers of standard deviations—set to 1.5. For large

gene families, where the consensus obtained contained a high proportion of gaps—i.e., where (consensus length − median length of sequences in the input) > 15% of median length of sequences, a second run of OD-Seq was performed. After the filtering of outlier protein sequences, the consensus was extracted from the multiple alignment using hmmemit in the HMMER3 package [120] (version 3.1b1, default parameters).

Then, OG consensus were aligned against each other using Blat [112] (version 36, default parameters), in order to detect unspecific regions, i.e., regions of 30 amino acids having a hit (with ≥ 85% identity) with another consensus (they typically correspond to common domains). The threshold of 85% was chosen because it corresponds to the average %identity observed when mapping reads from each coral species on consensus proteic sequences. Two thousand thirty-nine OG that contain unspecific domains were tagged. Additionally, transposon-like domains were looked for in the consensus and interproscan outputs from all genes in each OG, and 577 OG that were likely to correspond to transposable elements were also tagged. OG tagged as TE or unspecific domain-containing were not used for subsequent analyses.

### Construction of gene trees

Orthofinder provides gene trees for each OG, containing the 25 cnidarian species that were used as input. We also needed to generate trees for various subsets of species (for instance, only *Pocillopora* cf. *effusa*, only *Porites lobata*, only 11 coral species…). For each set of species needed, we extracted the corresponding proteic sequences after removal of outlier sequences (as described in " OG consensus construction" paragraph), and aligned them with MAFFT [126] (v7.464). Then we used FastTree [127] (2.1.11) with default parameters for construction of approximately maximum-likelihood phylogenetic trees. Trees were edited and visualized using Itol [128] and R ggtree package [129].

### Calculation of Ks between P. lobata and P. cf. effusa gene pairs

$K_s$ (rates of synonymous substitutions) were respectively calculated between pairs of *P. lobata* and *P.* cf. *effusa* paralogous genes after aligning protein sequences with muscle [124] (default parameters), and generating codon alignments with pal2nal [130] (V13). Then codeml (PAML package [131] version 4.8) was used to calculate dS values (i.e., $K_s$). The same procedure was used to calculate $K_s$ between the 2 allelic versions of the codings sequence (CDS) of each protein (Best Reciprocal Hits between haplotype 1 and haplotype 2) in both species.

### Mapping of short reads on OG consensus to estimate gene copy numbers

In order to estimate gene family copy number independently from assembly and annotation processes, we downloaded short-read datasets (Illumina paired-end) for 14 coral and 4 sea anemone species (Additional file 1: Table S7). We extracted 50 million sequences from pair1 files, trimmed to 100nt for consistency between analyses, and mapped those using diamond [121] on orthogroup coral consensus sequences. Unique hits were retained for each read, and depth of coverage was calculated on each consensus OG (25,210 OG after filtering TE and unspecific regions). Then the depth obtained for each OG was normalized for each species by dividing by the depth obtained on a set of conserved single-copy genes, in order for the final value obtained to be representative

of the gene copy number. Indeed, the ratio obtained for single-copy genes is close to 1 (Additional file 1: Figure S32A).

### Identification of a set of single-copy genes present in all corals

In order to normalize depths of mapping of short reads on each OG consensus, we needed a set of single-copy genes. We made a first attempt with BUSCO [107] version 4.0.2 (metazoa odb10 ancestral genes: 877 among the 954 consensus sequences were used, after discarding the ancestral genes that were not present in at least 10 coral species) but due to the divergence of corals with BUSCO metazoa ancestral sequences, few reads could be mapped and the depths obtained were low. Alternatively, we generated a set of genes that are present in exactly 1 copy in at least 13 coral species (among the 14 species listed in Additional file 1: Table S5). This conservative threshold avoids discarding genes that may have been missed or duplicated in the annotation of only one of the 14 genomes. The coral monocopy gene set contains 705 genes after removing OG with dubious transposon-related domains. For each species, after discarding a few outlier OG (with unexpectedly high or low coverages), we calculated the overall depth on the remaining single-copy OG. The values obtained were used to normalize the depths obtained by mapping the reads on each consensus OG. As expected, depths obtained after normalization on the set of 705 coral monocopy genes are tightly grouped around a value of 1. Contrastively, depths obtained on the same set of OG normalized with BUSCO metazoa are higher than 1 (due to the low coverage of mapping on BUSCO consensus), more heterogeneous between species (probably reflecting their distance to the consensus), and more variable inside species (Additional file 1: Figure S32). The set of 705 coral monocopy OG consensus could be used for other applications in coral comparative genomics. The consensus sequences are available in Additional file 13.

### Detection of amplified/reduced gene families between clades

To detect gene families with significantly expanded/reduced gene numbers between corals and sea anemones we calculated, for each OG (total = 25,210), the total number of genes in the groups of species to compare (11 coral species: *Fungia* sp.*, Orbicella faveolata, Goniastrea aspera, Stylophora pistillata, Pocillopora* cf. *effusa, Pocillopora verrucosa, Porites evermanni, Porites lutea, Porites lobata, Galaxea fascicularis, Acropora millepora*) vs 3 non-coral hexacorallia (*Aiptasia sp, Amplexidiscus fenestrafer, Discosoma sp*). We then performed a binomial test with a parameter of 11/14 to identify OG that display a proportion of coral genes that is significantly different from the expected proportion (11/14, under the null hypothesis where there are equal gene numbers in all species). Benjamini–Hochberg FDR correction for multiple testing was then applied and we selected the OG with corrected *P*-values < 0.001. The same procedure was performed to compare corals within the complex and robust clades, keeping only one species per genus (3 complex species: *Porites lobata, Galaxea fascicularis, Acropora millepora*, vs 5 robust species: *Fungia sp, Orbicella faveolata, Goniastrea aspera, Stylophora pistillata, Pocillopora* cf. *effusa*) with a parameter of 3/8 for the binomial test. Finally, to compare 4 massive (*Orbicella faveolata, Goniastrea aspera, Porites lobata, Galaxea fascicularis*) and 3 branched (*Acropora millepora, Pocillopora* cf. *effusa, Stylophora pistillata*) corals species, we used a parameter of 4/7. All calculations were performed with R (version 4.1.0).

### History reconstruction of gene family copy number variations

The CAFE5 software [58] was used to estimate gene copy numbers in internal nodes of the phylogenetic tree and identify branches with significant amplification or reductions of gene families. It was applied on the OG detected as amplified in corals in comparison to sea anemones, on the 11 corals and 3 sea anemone species. Among the 192 OG detected, 120 were present in the common ancestor of all species and could thus be analyzed with CAFE, but one (OG0000004) was removed because CAFE failed to identify parameters, probably because of the very large number of genes ($n = 3038$). After testing several sets of parameters, the following parameters were used: -p (poisson distribution for the root frequency distribution) -k 2 (number of gamma rate categories to use). *NB:* using the parameter -e (to estimate the global error model) provided almost identical results.

### Detection of enriched PFAM domains between clades

To assess putative differences in the proteome (amino acid translated genes) based on Pfam protein domains, genomic gene sets were annotated using InterProScan (v5.41–78.0 with default parameters). The following species were included in the analysis: Corals: *Acropora millepora* (Amil), *Porites lutea* (Plut), *Porites lobata* (Plob), *Porites evermanni* (Peve), *Galaxea fascicularis* (Gfas), *Stylophora pistillata* (Spis), *Pocillopora verrucosa* (Pver), *Pocillopora* cf. *effusa* (Pmea), *Fungia* sp. (Fungia), *Goniastrea aspera* (Gasp), *Orbicella faveolata* (Ofav). Others: *Aiptasia* sp. (Aiptasia), *Amplexidiscus fenestrafer* (Afen), *Discosoma* sp. (Disco). Based on the proportions for each Pfam domain per species (# pfam occurrences/# of all pfam occurrences for the genomic gene set), we determined standard deviations for each of the Pfam domains per species and performed non-parametric Mann–Whitney $U$ testing for each Pfam domain comparing pairs of groups. FDR was applied using qvalue [132] (version 2.24.0) for multiple test adjustment. Heatmaps were generated based on the top100 most significant domains for each comparison. Heatmaps were generated using the R package pheatmap [133] (version 1.0.12), with Pfam domain proportions being scaled across species by means of subtracting the overall mean (centering) and dividing by the overall standard deviation (scaling).

### Abundance quantification of Pocillopora cf. effusa tandem duplicated genes in meta-transcriptomic samples

Meta-transcriptomic reads of 103 samples coming from 11 islands were mapped on predicted transcript of *P.* cf. *effusa* using RSEM [134] (version 1.3.3, default parameters with bowtie2 option). This tool and its underlying model have been designed to properly take read mapping uncertainty into account, which is an important feature when dealing with duplicated genes. However, we checked for the potential impact of ambiguous read assignment to transcripts and found only 313 transcripts that contained no unique 31-mer and, on average, approximately 93% of reads from a sample were assigned uniquely to a given transcript. Uniq 31-mers were extracted using UniqueKMER [135], and quantifications analyses were performed using the TPM metric provided by RSEM. Pearson correlations of TPM distribution between all TDG were computed using the cor function of R (version 3.6.0) and the associated

*p*-value with rcorr function of Hmisc package of R. Heatmap showing the *z*-score of the mean TPM by OG was created using the heatmap.2 function of the R package gplots, and *z*-scores were computed with the scale function of heatmap.2. Data is provided in Additional file 14.

### Identification of innate immune system genes

#### *TIR-domain-containing orthogroups (putative Toll-like receptors, TLR)*

We annotated orthogroups (OG) where InterProScan detected TIR domains (IPR000157, IPR035897) in at least 20% of the genes in 14 cnidarian species, as TIR-domain-containing orthogroups. We obtained 21 OG totalizing 643 genes in 14 cnidaria species. The threshold (20% of genes containing the domain required to annotate the OG) was set by inspection of a manually curated set of TIR-containing OG. In *Porites lobata*, and *Pocillopora* cf. *effusa*, respectively 56 and 47 genes belong to the identified OG, among which 49 (87.5%) and 42 (89.4%) contain the TIR domain. The relatively low threshold is due to more difficult domain identification in other species, where gene annotations can be fragmented.

#### *NACHT domain-containing orthogroups (putative NOD-like receptors, NLR)*

We annotated OG where InterProScan detected NACHT domain (IPR007111) in at least 5% of the genes as putative NLR orthogroups. We obtained 46 OG totalizing 2991 genes in 14 cnidaria species. As described above, the thresholds were set by inspection of a manually curated gene set, and the % of genes in retained OG that actually contain the NACHT domain is high in *P. lobata* and *P.* cf. *effusa*.

#### *Lectin domain-containing orthogroups*

We annotated OG where InterProScan detected lectin-like domains (IPR001304, IPR016186, IPR016187, IPR018378, IPR019019, IPR033989, IPR037221, IPR042808) in at least 50% of the genes as putative lectin orthogroups. We obtained 81 OG totalizing 2475 genes in 14 cnidaria species.

#### *Domain composition*

We studied the domain composition of TIR and NACHT/NB-ARC containing proteins in 5 species: *A. digitifera*, *P. lobata*, *P. lutea*, *P.* cf. *effusa*, and *P. verrucosa*. We used all genes containing TIR (IPR000157, IPR035897) and NACHT/NB-ARC (IPR007111/IPR002182) domains (not only the ones in OG fulfilling the criteria mentioned above). We derived domain compositions from InterProScan outputs after manual curation to discard redundant domains. The list of genes and domain compositions are available in Additional files 15 and 16. For NACHT/NB-ARC-containing proteins, we identified truncated proteins when less than 250 aa were annotated and no domain was detected with InterProScan upstream and/or downstream of the NACHT/NB-ARC domain. When the upstream/downstream sequence contained more than 250 aa, and no domain was annotated, the gene was tagged as "noneDetected." Simplified domain compositions obtained are displayed in Additional file 17.

## Supplementary Information

**Additional file 1: Table S1.** Statistics of the ONT sequencing data. **Table S2.** Raw long-read assemblies. A. Porites lobata. B. Pocillopora cf. effusa. **Table S3.** Statistics of all assemblies. **Table S4.** Statistics of gene prediction on reference and alternative haplotype assemblies. **Table S5.** BUSCO analyses of raw and haploid assemblies. **Table S6.** Repeat composition of coral assemblies. **Table S7.** Data used for comparison of 26 Cnidaria species. **Table S8.** Coverage of genome assemblies by syntenic blocks (for contigs with at least 5 genes). **Table S9.** Validation of pairs of adjacent duplicate genes and clusters of tandemly duplicated genes. **Table S10.** Table of calcification-related proteins in non-scleractinian and scleractinian - robust and - complex coral species. **Figure S1.** K-mer distribution and estimation of genome sizes and heterozygous rates from short reads. **Figure S2.** KAT plot of Porites lobata assemblies. **Figure S3.** KAT plot of Pocillopora cf. effusa assemblies. **Figure S4.** K-mer profiling for all assemblies of Porites lobata and Pocillopora cf. effusa **Figure S5.** Distribution of the number of genes in tandemly repeated gene clusters. **Figure S6.** Validation of pairs of adjacent duplicated genes and clusters of TDG. **Figure S7.** Validation of pairs of adjacent duplicated genes. **Figure S8.** Validation of TDG clusters. **Figure S9.** Comparison of haplotype assemblies and annotations for a TDG cluster. **Figure S10.** Two P. lobata genomic regions that illustrate proteins and transcripts alignments before or after the adaptation of gene prediction workflow for tandemly duplicated genes. **Figure S11.** Consensus sequences of palindromic ITS satellites, found in Porites lobata, Porites evermanni and Porites lutea genomes. **Figure S12.** Conservation of orthologous genes in Porites lobata and Pocillopora cf. effusa with other Cnidarian. **Figure S13.** Number of TDG genes vs total number of genes on contigs. **Figure S14.** Expanded PFAM domains in corals vs non-corals hexacoralia. **Figure S15.** OGs amplified in complex vs robust corals. **Figure S16.** Expanded PFAM domains in complex vs robust corals. **Figure S17.** OGs amplified in massive vs branched corals. **Figure S18.** Heatmap representing the *P*-values obtained with CAFE for gene number amplification and reduction events for each node on the phylogenetic tree. **Figure S19.** Gene family amplifications/reductions identified with CAFE. **Figure S20.** Distribution of Ks between TDG gene pairs in four classes of intergenic distances. **Figure S21.** View of the Pocillopora cf. effusa genome browser, with 2 adjacent genes duplicated in tandem. **Figure S22.** Distribution of Ks between TDG pairs in 3 structural conservation classes. **Figure S23.** View of a cluster of 3 tandemly duplicated genes belonging to OG0000023 (lectin-like domain). **Figure S24.** Expression quantification of 200 random OGs across the 103 meta-transcriptome samples. **Figure S25.** Number of genes in OG identified as TLR, NLR and lectins. **Figure S26.** Percent of NACHT/NB-ARC containing proteins that are likely truncated at the N-terminal, C-terminal or both extremities. **Figure S27.** Observed N-terminal and C-terminal domains for NACHT/NB-ARC containing proteins in five coral species. **Figure S28.** Graphic representation of the number of calcification-related proteins in non-scleractinian and scleractinian species and associated *p*-values. **Figure S29.** Maximum likelihood phylogenetic tree of cnidarian ammonium transporters Amt1. **Figure S30.** Genomic localisation of cnidarian ammonium transporters Amt1 in various species. **Figure S31.** Genomic localisation of Bicarbonate Anion Transporter SLC4 in various species. **Figure S32.** Coral monocopy gene set.

**Additional file 2.** OG significantly amplified in corals vs sea anemones.

**Additional file 3.** The top100 Pfam domains exhibiting most significant differential abundance between cnidaria and scleractinian corals.

**Additional file 4.** Describing OG significantly amplified in robust vs complex clades.

**Additional file 5.** Describing OG significantly amplified in complex vs robust clades.

**Additional file 6.** Describing the top100 Pfam domains exhibiting most significant differential abundance between robust and complex clade corals.

**Additional file 7.** Describing OG significantly amplified in massive vs branched corals.

**Additional file 8.** Describing OG significantly amplified in branched vs massive corals.

**Additional file 9.** Providing number of genes annotated in TIR containing, NACHT containing and lectin containing OGs.

**Additional file 10.** Describing the 5 most abundant domains (IPR) identified with interProScan on 25 Cnidarian species, for each OG.

**Additional file 11.** Describing the 5 most abundant best BlastP Hits on nrprot for Pocillopora cf. effusa proteins in each OG.

**Additional file 12.** Describing the 5 most abundant best BlastP Hits on nrprot for Porites lobata proteins in each OG.

**Additional file 13.** Fasta file providing consensus sequences of 705 monocopy OGs found in all corals.

**Additional file 14.** Containing the heatmap data used for Figure 7C, (Z-scores of mean TPM per OG: rows= genes (OG), columns=Tara Pacific Islands).

**Additional file 15.** ProvidingTIR containing genes domain composition.

**Additional file 16.** Providing NACHT/NB-ARC containing genes domain composition.

**Additional file 17.** Providing NACHT/NB-ARC containing genes with description of N-terminal, central and C-terminal domains.

**Additional file 18.** Review history.

## Acknowledgements

Special thanks to the Tara Ocean Foundation, the R/V Tara crew and the Tara Pacific Expedition Participants (https://doi.org/10.5281/zenodo.3777760). We are keen to thank the commitment of the following institutions for their financial and scientific support that made this unique Tara Pacific Expedition possible: CNRS, PSL, CSM, EPHE, Genoscope, CEA, Inserm, Université Côte d'Azur, ANR, agnès b., UNESCO-IOC, the Veolia Foundation, the Prince Albert II de Monaco Foundation, Région Bretagne, Billerudkorsnas, AmerisourceBergen Company, Lorient Agglomération, Oceans by Disney, L'Oréal, Biotherm, France Collectivités, Fonds Français pour l'Environnement Mondial (FFEM), Etienne Bourgois, and the Tara Ocean Foundation teams. This is publication number 22 of the Tara Pacific Consortium.

## Review history

The review history is available as Additional file 18.

## Peer review information

## Authors' contributions

JP, EB, VB, SP, and DZ managed the sampling of coral material. JP, DZ, KL, and CC performed DNA/RNA extractions. KL, CC, and JMA coordinated DNA/RNA sequencing, AC and JMA produced the assemblies. BN and CDS performed the annotation of the genomes. AR, MP, and EG carried out analyses of telomeric repeats. CBL and CRV performed the comparative analysis based on pfam domains. LC, ST, and DZ carried out the analyses on calcification-related genes. EA, QC, and JLH provided help on the analysis of transcriptomic data from environmental samples. BN, FD, and JMA performed the genomic analyses featured in this manuscript. EB, CM, GB, GI, SRo, EG, DZ, PW, and CRV coordinated the Tara Pacific scientific expedition. SA, DA, BB, EB, EB, CB, CdV, ED, MF, DF, PF, PG, EG, FL, SPe, SPl, SRe, MBS, SS, OT, RT, RVT, CRV, PW, and DZ are coordinators of the Tara Pacific consortium. DA and SP are the scientific directors of the Tara Pacific expedition. All authors have read and approved the final version of the manuscript.

## Authors' Twitter handles

Twitter handles: @BeNoel7 (Benjamin Noel), @alice_rouan (Alice Rouan), @CarolBuitragoL7 (Carol Buitrago-López), @EricJ_Armstrong (Eric Armstrong), @QuentinCarradec (Quentin Carradec), @juleye91 (Julie Lê-Hoang), @reefgenomics (Christian R Voolstra), @J_M_Aury (Jean-Marc Aury).

## Funding

This work was supported by the Genoscope, the Commissariat à l'Énergie Atomique et aux Énergies Alternatives (CEA) and France Génomique (ANR-10-INBS-09–08).

## Availability of data and materials

The Illumina and PromethION sequencing data are available in the European Nucleotide Archive under the following project PRJEB51539 [136]. Genome assemblies and gene predictions are freely available from the Genoscope website [137]. Additionally, all data and scripts used to produce the main figures are available under CeCILL license on a github repository [138] and through Zenodo [139].

# Declarations

## Ethics approval and consent to participate

Not applicable.

## Competing interests

The authors declare that they have no competing interests. JMA received travel and accommodation expenses to speak at Oxford Nanopore Technologies conferences.

## Author details

[1]Génomique Métabolique, Genoscope, Institut François Jacob, CEA, CNRS, Univ Evry, Université Paris-Saclay, Evry 91057, France. [2]Research Federation for the Study of Global Ocean Systems Ecology and Evolution, R2022/Tara Oceans GO-SEE, 3 Rue Michel-Ange, 75016 Paris, France. [3]Université Côte d'Azur, CNRS, Inserm, IRCAN, Nice, France. [4]LIA ROPSE, Laboratoire International Associé, Université Côte d'Azur - Centre Scientifique de Monaco, France. [5]Department of Biology, University of Konstanz, Constance, Germany. [6]Centre Scientifique de Monaco, Marine Biology Department, Monaco City 98000, Monaco. [7]Sorbonne Université, Collège Doctoral, 75005 Paris, France. [8]Laboratoire d'Excellence CORAIL, PSL Research University, EPHE-UPVD-CNRS, USR 3278 CRIOBE, Université de Perpignan, 52 Avenue Paul Alduy, 66860, Cedex Perpignan, France. [9]Genoscope, Institut François Jacob, CEA, CNRS, Univ Evry, Université Paris-Saclay, Evry 91057, France. [10]Fondation Tara Océan, Base Tara, 8 Rue de Prague, 75 012 Paris, France. [11]School of Marine Sciences, University of Maine, Orono, USA. [12]AD2M, UMR 7144, Sorbonne Université, CNRS, Station Biologique de Roscoff, ECOMAP, Roscoff, France. [13]Shimoda Marine Research Center, University of Tsukuba, 5-10-1, Shimoda, Shizuoka, Japan. [14]Institut de Biologie de L'Ecole Normale Supérieure (IBENS), Ecole Normale Supérieure, CNRS, INSERM, Université PSL, 75005 Paris, France. [15]Laboratoire Des Sciences du Climat Et de L'Environnement, LSCE/IPSL, CEA-CNRS-UVSQ, Université Paris-Saclay, Gif-Sur-Yvette 91191, France. [16]Department of Earth and Planetary Sciences, Weizmann Institute of Science, 76100 Rehovot, Israel. [17]Sorbonne Université, CNRS, Laboratoire d'Ecogéochimie des Environnements Benthiques (LECOB), Observatoire Océanologique de Banyuls, Banyuls Sur Mer, France. [18]Institut de La Mer de Villefranche Sur Mer, Sorbonne Université, Laboratoire d'Océanographie de Villefranche, Villefranche-Sur-Mer 06230, France. [19]Institut Universitaire de France, Paris 75231, France. [20]European Molecular Biology Laboratory, European Bioinformatics Institute, Wellcome Genome Campus, Hinxton, Cambridge CB10 1SD, UK. [21]Departments of Microbiology and Civil, Environmental and Geodetic Engineering, The Ohio State University, Columbus, OH 43210, USA. [22]Department of Biology, Institute

of Microbiology and Swiss Institute of Bioinformatics, ETH Zürich, Vladimir-Prelog-Weg 4, CH-8093 Zurich, Switzerland. [23]School of Biological and Chemical Sciences, Ryan Institute, University of Galway, University Road H91 TK33, Galway, Ireland. [24]Department of Microbiology, Oregon State University, 220 Nash Hall, Corvallis, OR 97331, USA. [25]Department of Human Genetics, CHU Nice, Nice, France.

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

## 