## [**Additional file 18. **Review history. · Genome Biology]

Review History

First round of review

Reviewer 1

Are you able to assess all statistics in the manuscript, including the appropriateness of statistical tests used? No, I do not feel adequately qualified to assess the statistics.

Comments to author:

The authors sequenced genomes of two corals *Porites lobata* and *Pocillopora meandrina* using ONT long reads and Illumina short reads from the colony samples. In addition, the genome assembly of the coral *Porites evermanni* was obtained using Illumina short reads. They annotated and compared their genomes with published coral genomes. It is found that the two genomes from long reads had more predicted genes than short reads because it is difficult to assemble and annotate coral genomes with a high number of tandem duplicated genes using short reads. They focused on detailed analysis of duplicated genes that had occurred in the coral lineage. The analysis showed the importance of previously missed duplicated genes to inform the biology of reef-building corals. Results of comparative analyses are well shown but evaluations for genome assemblies and expression analysis are insufficient. Tandem duplication and convergent evolution in corals are discussed well. However, some explanations that are logically incorrect are found. Therefore, I cannot strongly recommend the publication of this manuscript in *Genome Biology*.

Major concerns:

Page 1, 16-20: The explanation for higher gene number that is due to short-read sequencing technologies is insufficient. It is difficult to understand it without comparisons between short-read assembly and long-read assembly that are obtained from the same sample. Higher gene numbers in *Porites lobata* and *Pocillopora meandrina* are likely to be explained by the method to generate a haploid version and by lineage specific duplications. Because cumulative genome sizes are larger than estimated genome sizes probably by high genome heterozygosity (646 Mb/543 Mb and 347 Mb/315 Mb in Table 1), the analysis for the explanation of the difference will be needed. Figs. S2B and S3B support that heterozygous content is found in haploid version. The difference of coverage by read mapping to contigs may estimate possible heterozygous content in haploid assemblies. Since the *Porites evermanni* genome with large number of predicted genes has been assembled using short reads, the abstract does not correspond to the result.

Page 1, 20-28: It includes inaccurate conclusion. Published coral genome sequences have revealed tandem duplicated genes related to immunity (e.g., Hamada et al. 2013). Diversification of immune genes has been discussed in comparative analysis of the *Pocillopora damicornis* genome (Cunning et al., 2018). Because the point has been discussed repetitively, the importance of duplicated genes is not missed in the previous papers.

Page 10, 1-9: The expression analysis in Fig. 7 includes high novelty but is not shown with control data. It may be difficult to investigate the expression dosage of duplicated genes using the environmental samples. However, it will be possible to analyze it using transcriptomes from Moorea samples that were also used for genome sequencing. Comparisons with control

experiments are needed for the interpretation of the environmental expression data.

Specific comments:

Page 3, 8-10 and Page 14, 8-20: Longevity features of *Porites* are interesting. But two different speculations without longevity related data are described. Comparative data of coral longevity may be shown in Fig. 1D and Fig. 3A.

Page 4, 33-39: The *Acropora millepolla* genome has been used for comparative analysis at the chromosomal level (Cooke et al. 2020). The assembly without chromosomal information is not the most complete coral genome.

Page 15, 23-25: The authors used genomic DNAs from frozen tissues of coral colonies that probably included Symbiodiniaceae and bacteria (Page 1, 42-48). However, the method for deleting bacterial DNA is not shown.

Page 19, 40-41, *Fungia* spp. => *Fungia* sp.

Fig. 1C: Scaffold N50 or Contig N50 are shown.

Fig. 1F: The data for *Pocillopora acuta* is not shown.

Table S1: "Genome size of 650Mb" and "Genome size of 350Mb" look wrong. Both do not correspond to the data in Table 1.

Fig. S4: Many transcript variants are predicted in the genomic regions. However, the number of variants in predicted genes is not shown.

Reviewer 2

Are you able to assess all statistics in the manuscript, including the appropriateness of statistical tests used? There are no statistics in the manuscript.

Comments to author:

It is an extensive comparative research on coral assemblies.

This manuscript is one of the most extensive assembly/genome comparison reports I have seen for coral genomes. The amounts of data generated for the genomes are large and the quality of data is high with ONT long read sequencing on top of short read sequencing data.

I wonder why they did not use Hi-C long distance mapping sequencing.

1. The title: "Pervasive gene duplications as a major evolutionary driver of coral biology" is too obvious and plain to me. Gene duplication as the driver of genome evolution has been well established since the first bacterial genomes (even before, it was known but confirmed with hard evidence with three complete bacterial genomes). This reviewer recommend it to be a bit more specific. Such as adding duplicated gene families. "coral biology" is too general. If it was to do with some kind of resistance to environmental

insults, it can be said at the title level. Also, my guess is that previous genome assembly studies must have shown data supporting it.

Also, it is not really novel or surprising. Is it? Just reading the title, I expected it to be some kind of extensive gene duplication analyses, but the paper is a small book of comparing newly constructed high quality coral genomes with existing ones, with all kinds of typical analyses.

Also, the paper is about "tandemly" duplicated genes rather than gene duplication itself.

Abstract:

2. "a relatively higher gene number" in abstract. It will be good to denote the actual numbers. (forcing readers to sift through numbers... while it is the foundation of the research problem).

3. "a high number of tandemly duplicated genes, ". Again. This is a genome paper and I, as a reader, want to know how many such TDGs are in the coral assemblies as soon as possible. That is the point of abstract anyway.

4. "which are generally difficult to assemble and annotate using short-read sequencing technologies". This is not necessary in abstract. Wasting the word count...

5. "These duplicated genes, " At least some gene families will make readers to know what "these" really look like.

Introduction:

7. "We posit that gene duplication is an evolutionary driver of coral biology that contributes to their longevity"

 Gene duplication is always the driver of genome evolution. That is why we have almost 1Gbp coral genomes now. I.e., it is too obvious. It is not a discovery. In this case, the gene duplication of certain families is the point, IMO. Many previous studies have checked and shown gene expansion and contraction. If there is any novel TDG(s) or TDG mechanism in corals, it would be interesting. I suggest it to be removed or merged with the previous already very long sentence: "Our results expose the vast presence of duplicated gene families in both coral genomes mapping to functions associated with the innate immune system, which escaped previous analyses based on fragmented and incomplete genomes assemblies due to sequencing method constraints."

8. "The *P. lobata* and *P. meandrina* genomes were sequenced using a combination of ONT long reads and Illumina short reads"

 No. Rephrase it. The two genomes were not sequenced by a combination of ONT and Ill... reads. By ONT and Illumina "sequencers", "machines" or "platforms".

9. "and a high level of heterozygosity was detected, 2.3% and 1.14% respectively"

 2.3% is high for a coral? Corals and marine species like that contain quite high heterozygosity as far as I know. High compared to what?

10. "The resulting haploid assemblies, with a cumulative size of 646 Mb for *P. lobata* and 347 Mb for *P. meandrina*, contained...."

 with cumulative size"s" of *P. lobata* and ... *P. meandrina*...

11. "we sequenced the genome of *Porites evermanni* using short-reads."

 Again, you did not sequence it with short-reads. You sequenced it with short-read Illumina sequencers.

12 "Although more fragmented,"

 Although much fragmented? Or Although more fragmented than XXX, ..

13. "These particular genes"

 Particular? It would be clear to say These TDGs...

14. "the annotation of these TDG"

 I wish those TDGs were stated as early as possible. The paragraph is depicting what the authors have done. It would be better to put forward the results first.

The contribution is such a new assembly protocol that successfully found TDGs and telling what the TDGs as early as possible will make readers to keep paying attention.
(Anyway, this is not critical)

15. "Noteworthy, we noticed in the three Porites species"

 Rephrase this. Noteworthy.. noticed...

16. "Attempts to search for this sequence failed to find it outside the Porites genus"

 It is not natural. Search for.... failed and again to "find it". Just "Attempts to search for this sequence failed outside the Porites genus" will be fine.

17. "true duplications from allelic copies of a given gene."

 ...duplicates?

18. "To further compare coral genomes, orthologs with 22 other cnidarian species were calculated"

 What do you mean by "calculated"? analyzed?

19. "Notably, although an initial matter of debate,"

 Notably, as an initial matter/point for debate? Rephrasing will be good.

Reviewer 3

Are you able to assess all statistics in the manuscript, including the appropriateness of statistical tests used? Yes, and I have assessed the statistics in my report.

Comments to author:

The manuscript by Noel et al. 2022 focuses on the generation and analysis of two new coral genomes (*Porites lobata* and *Pocillopora meandrina*) sequenced with Oxford Nanopore technology. After performing gene prediction on the assemblies, the authors found a significantly higher number of predicted genes in these genomes than in other coral (and other non-coral cnidarian) genomes and explain this by identifying a high number of tandemly duplicated genes and make claims about why certain gene families may have been duplicated in these lineages.

Although the authors have done an admirable job in generating these important genome assemblies and made an attempt to try to detect allelic duplications in their assemblies and perform haplotype phasing using Haplomerger2 to generate haploid versions of their assemblies, I am not fully convinced that this process has successfully phased their assemblies and fully removed "false duplications" from their haploid assemblies. Unfortunately, if false duplications remain, it can negatively affect all downstream analyses. Therefore, interpretations of the data and the evidence supporting the main findings of this study that there are a high number of tandemly duplicated genes in two coral genomes may be weak.

I give my reasoning and evidence to support this line of thinking below.

As noted in Guan et al. 2020 (Bioinformatics 36(9) 2896- 28980), several tools have been developed to try to resolve the problem of "false duplication" in genome assemblies and Haplomerger2 is one such tool. The main problem with Haplomerger2 - as pointed out in this paper - is that it ignores read depth and relies only on the alignment of contigs to each other. The authors of that paper suggest that Haplomerger2 can make false joins to link adjacent contigs leading to false duplications, among other issues, and suggest trying their tool, "purge_dups", which does account for read depth, to overcome these issues.

Additionally, a preprint by Ko et al. 2021 (<https://doi.org/10.1101/2021.04.09.438957>) goes into great depth on the topic of how false duplications in genome assemblies lead to false biological conclusions. Falsely duplicated genomic regions and genes can lead to overestimated gene family expansions. The main source of this occurs where haplotype sequences are more divergent than other parts of the genome which can lead assembly algorithms to classify them as separate regions or genes. Ko et al. discuss how better methods are still needed to better separate haplotypes and urge extremely cautious analysis on gene gains. Without proper phasing/haplotype separation, both alleles of heterozygous loci can be assembled into one scaffold (or two different scaffolds). Ko et al. use two independent methods to try to tackle these issues, including Cactus for whole-genome alignment between two assemblies (they were comparing different assemblies) and "purge_dups" for self-alignment.

I agree with the points made by Guan et al. and Ko et al. and urge Noel et al. to be extremely cautious in their interpretation of their current assemblies. I believe that more testing and careful analysis needs to be done to determine if they still have "false duplications" present in their haploid assemblies before moving forward with their analyses of the genomes. My main reasons for concern stem from three pieces of information presented in the manuscript (1) these corals have very high estimated levels of heterozygosity across their genomes (*P. lobata* estimated at 2.3% and *P. meandrina* estimated at 1.14%), (2) the fact that their haploid assembly sizes are significantly larger than their estimated genome sizes (*P. lobata* estimated genome size 543 Mb, final assembly size is 646 Mb; *P. meandrina* estimated genome size 315 Mb, final assembly size is 347 Mb) and (3) their predicted gene numbers are much higher than other cnidarian genomes (*P. lobata* has 42,872 predicted genes, *P. meandrina* has 32,095 predicted genes).

To me, these three statistics amount to red flags that provide ample evidence that strongly indicates that the genome assembler they chose (SMARTdenovo) as well as their Haplomerger2 analysis were unable to completely resolve the two haplotypes and they have retained some false duplications in their haploid assemblies, especially in *P. lobata*. The authors use BUSCO and KAT analysis to show a reduction in allelic duplications, but this reduction does not appear to be "complete" which could lead to false duplications remaining in some areas of the genome, particularly the regions that appear to contain tandemly duplicated genes. Even the Ks distribution data in Figure 5 for the tandemly duplicated gene pairs seems to support the idea that these could be false duplications, likely retained due to the fact that the phasing was incomplete due to high heterozygosity in these regions.

At the very least, I would urge the authors to run 'purge_dups' on their assemblies and examine how their resulting primary haploid assemblies compare with their current assemblies from

Haplomerger2. I would also recommend running some k-mer profiling analyses on the resulting assemblies after running 'purge_dups', like in Extended Data Fig. 2 of Ko et al. 2021. There are also assemblers, like NextDenovo, that have been shown to perform better on more repetitive and highly heterozygous genomes (e.g. Sun et al. 2021 <https://doi.org/10.1098/rstb.2020.0160>) that the authors could try. It will be very important to perform more careful analysis of their haplotype assemblies to identify and remove false duplications before making claims about gene gains and tandemly duplicated genes in these corals.

Since I have these concerns about the underlying assemblies, I will limit my comments to this aspect of the manuscript as it can affect nearly all the analyses and conclusions that follow. As Ko et al. (2021) point out, even though long-read sequencing is better at resolving repetitive regions of genomes, long reads are not able to fully resolve false duplications and I suspect that this manuscript may suffer from a significant proportion of false allelic duplications remaining in their assemblies, obscuring the authors' ability to distinguish between true duplications and false duplications and to make well-supported biological conclusions.

I do think these coral genomes make an important contribution to cnidarian genomics, and in particular, coral genomics, and I found the overall writing and methods used in general to be clear and reasonable. This problem of false allelic duplication is a very tricky and complicated one that has not yet been solved, particularly for genomes with high levels of heterozygosity - even for vertebrate genomes - and I hope the authors will not take my comments as a condemnation of their science but rather an attempt to provide important up-to-date information on this very real problem and an opportunity to use additional tools to better assess if what they are seeing with the tandemly duplicated genes is actually "real" or not. I would be happy to take another look at the manuscript after the authors perform additional analyses and provide more evidence that they have gone further to address the problem of false duplication in their data.

First of all, we would like to thank the three reviewers for their meticulous reading of our article. We have taken into account their comments. To make the review process easier, all the changes we made are in blue in the main article and supplementary data.

Reviewer #1:

The authors sequenced genomes of two corals *Porites lobata* and *Pocillopora meandrina* using ONT long reads and Illumina short reads from the colony samples. In addition, the genome assembly of the coral *Porites evermanni* was obtained using Illumina short reads. They annotated and compared their genomes with published coral genomes. It is found that the two genomes from long reads had more predicted genes than short reads because it is difficult to assemble and annotate coral genomes with a high number of tandem duplicated genes using short reads. They focused on detailed analysis of duplicated genes that had occurred in the coral lineage. The analysis showed the importance of previously missed duplicated genes to inform the biology of reef-building corals. Results of comparative analyses are well shown but evaluations for genome assemblies and expression analysis are insufficient. Tandem duplication and convergent evolution in corals are discussed well. However, some explanations that are logically incorrect are found. Therefore, I cannot strongly recommend the publication of this manuscript in *Genome Biology*.

Major concerns:

Page 1, 16-20: The explanation for higher gene number that is due to short-read sequencing technologies is insufficient. It is difficult to understand it without comparisons between short-read assembly and long-read assembly that are obtained from the same sample. Higher gene numbers in *Porites lobata* and *Pocillopora meandrina* are likely to be explained by the method to generate a haploid version and by lineage specific duplications.

Indeed, we combined long-read sequencing and a dedicated gene prediction method. We also expect lineage-specific gene duplications, but we have shown using short-read alignment on orthogroups that existing assemblies (especially *Porites lutea*, *Pocillopora damicornis* and *Pocillopora acuta*) lack duplicated genes. This could be due to either assembly or gene prediction methods. Notably, previous coral genome studies have shown that corals exhibit high levels of heterozygosity, which makes resolving haplotypes difficult. There is a large number of bioinformatics software with their own heuristics and biases, which make it highly challenging to compare short- and long-read based assemblies even using data from the same sample. However, in our analysis, we compared several assemblies and annotations that were generated using a wide variety of methods and showed that the vast majority of existing coral genome assemblies are not complete at the gene level. As reported by reviewer 3, we have added several analyses to validate our haploid assembly and clusters of tandemly duplicated genes.

Because cumulative genome sizes are larger than estimated genome sizes probably by high genome heterozygosity (646 Mb/543 Mb and 347 Mb/315 Mb in Table 1), the analysis for the explanation of the difference will be needed.

We agree with the reviewer that there is a difference between estimated genome sizes and assembly sizes. However, genome sizes have been estimated using a k-mer approach which is sometimes not fully accurate, due to heterozygosity and possibly the presence of several other organisms. In our case, we found a few contigs that correspond to the mitochondrial genome of symbiotic algae. Moreover, we used Haplomerger2 to generate haploid

assemblies, and obtained two haplotypes for each genome. A reference version composed of the longer haplotype (when two haplotypes are available for a genomic locus) and a second version, named alternative, with the corresponding other allele of each genomic locus. Consistently keeping the longest allele in the reference haplotype explains the larger size of the reference assembly (for *Porites lobata*, 642Mb and 588Mb for reference and alternative assemblies respectively). This information was already mentioned in the method section, but we have added more details to make it clearer, in the following method section: “Reconstruction of allelic relationships and haploid assembly”.

Figs. S2B and S3B support that heterozygous content is found in haploid version. The difference of coverage by read mapping to contigs may estimate possible heterozygous content in haploid assemblies.

It is certain that in a haploid version of the genome we expect to find both heterozygous (peak at 45X on Figure S2B) and homozygous regions (peak at 90X on Figure S2B). Our main objective was to avoid keeping allelic duplications in the haploid assembly, i.e. regions where both alleles of a given region are still present. It was achieved as shown by the small area of the purple curve in the KAT plots of Figure S2B and S3B, and also by the low number of duplicated genes in the BUSCO results.

Since the *Porites evermanni* genome with large number of predicted genes has been assembled using short reads, the abstract does not correspond to the result.

Indeed *Porites evermanni* genome assembly was obtained using short reads but gene prediction was performed using our dedicated method which allows better retrieval of duplicated genes. Genome assembly and gene prediction methods both have an impact. We modified the abstract and removed the following information: “which are generally difficult to assemble and annotate using short-read sequencing technologies”.

Page 1, 20-28: It includes inaccurate conclusion. Published coral genome sequences have revealed tandem duplicated genes related to immunity (e.g., Hamada et al. 2013). Diversification of immune genes has been discussed in comparative analysis of the *Pocillopora damicornis* genome (Cunning et al., 2018). Because the point has been discussed repetitively, the importance of duplicated genes is not missed in the previous papers.

We agree with the reviewer and temper our conclusion by removing “previously missed”. Indeed, gene duplications have been revealed in previous publications, at least for one specific gene family (NACHT/NB-ARC genes) and the authors suggested that this gene family was generated in part by tandem duplication (Hamada et al. 2013). However, pervasive tandem duplications and the huge number of duplicated genes in all corals have not been discussed previously.

Page 10, 1-9: The expression analysis in Fig. 7 includes high novelty but is not shown with control data. It may be difficult to investigate the expression dosage of duplicated genes using the environmental samples. However, it will be possible to analyze it using transcriptomes from Moorea samples that were also used for genome sequencing. Comparisons with control experiments are needed for the interpretation of the environmental expression data.

When the DNA of *Pocillopora meandrina* was extracted, we also extracted RNA and sequenced this sample which comes from the same individual. However, we only get a single sample, which is not sufficient to investigate gene dosage. We added this sample in Figure 7C. Importantly, even if the temptation to use Moorea samples was high, it is important to

mention that these samples were collected during the Tara Pacific expedition and not from the same site as the sample used for genome sequencing. Obviously, as for other samples we could not exclude a difference in the number of copies between colonies. In addition, gene expression through sampling sites was investigated in another manuscript (<https://doi.org/10.1101/2021.11.12.468330>).

Specific comments:

Page 3, 8-10 and Page 14, 8-20: Longevity features of *Porites* are interesting. But two different speculations without longevity related data are described. Comparative data of coral longevity may be shown in Fig. 1D and Fig. 3A.

We have added information on longevity in the relevant paragraphs. Indeed, several 1000-year-old colonies of *Porites lobata* have already been reported. However, these data are difficult to interpret, as they are an observation of age (recorded maximum ages of corals, Bythell, J. C., et al 2018) and may be an underestimate of longevity. Since only four species that have been studied here have age records, we chose not to include this data in Figures 1 and 3.

Page 4, 33-39: The *Acropora millepora* genome has been used for comparative analysis at the chromosomal level (Cooke et al. 2020). The assembly without chromosomal information is not the most complete coral genome.

When we started the comparative analysis and the building of orthogroups, the chromosome-scale assembly of *Acropora millepora* was not available, and we used the version of Ying et al. (2019). Before the submission of our article, we incorporated the version by Fuller et al. (2020) in Figure 1, which had a good contiguity but at that time a poor gene prediction (version 2.01). A new gene prediction (2.1) was released later (Oct 14 2021, https://www.ncbi.nlm.nih.gov/genome/annotation_euk/Acropora_millepora/101/). We now have updated Figure 1 to reflect this new version of the annotation (except panels E and F that were generated using assembly from Ying et al, as functional annotations and orthogroups were calculated on this version), and made the information on versions used available in Table S7.

Page 15, 23-25: The authors used genomic DNAs from frozen tissues of coral colonies that probably included Symbiodiniaceae and bacteria (Page 1, 42-48). However, the method for deleting bacterial DNA is not shown.

We have added a paragraph in the Methods section ("Contamination removal"). As we used a DNA extraction method based on a nuclei isolation approach, we minimized contamination by Symbiodiniaceae. We found few contigs corresponding to the mitochondrial genome of Symbiodiniaceae and to bacterial genomes.

Page 19, 40-41, *Fungia* spp. => *Fungia* sp.

We have modified the text accordingly.

Fig. 1C: Scaffold N50 or Contig N50 are shown.

We initially used both depending on the sequencing strategy, but we modified Figure 1C and accounted for the contig N50 value which is common to all assemblies (scaffolds are broken at each unknown base).

Fig. 1F: The data for *Pocillopora acuta* is not shown.

Indeed, we did not use the *Pocillopora acuta* gene prediction in our comparative analysis, because its number of annotated genes was higher than expected based on comparison to other corals and only a small proportion contained domains. It is likely due to the fact that the authors merged transcripts predicted with RNA-Seq data and transcripts from *ab initio* predictions. This information has been added to the methods section.

Table S1: "Genome size of 650Mb" and "Genome size of 350Mb" look wrong. Both do not correspond to the data in Table 1.

In Table S1, we believe it is important to show coverage relative to genome assembly sizes and prefer to use these values. We corrected "Genome size of xxxMb" to "Assembly size of xxxMb".

Fig. S4: Many transcript variants are predicted in the genomic regions. However, the number of variants in predicted genes is not shown.

Indeed, we did not predict transcript isoforms, our method generates a single transcript for each gene.

Reviewer #2:

It is an extensive comparative research on coral assemblies.

This manuscript is one of the most extensive assembly/genome comparison reports I have seen for coral genomes. The amounts of data generated for the genomes are large and the quality of data is high with ONT long read sequencing on top of short read sequencing data.

I wonder why they did not use Hi-C long distance mapping sequencing.

Unfortunately, the material requirement for long-read sequencing did not allow us to perform Hi-C libraries on the same sample. We were only able to produce long- and short-reads from the collected samples.

1. The title: "Pervasive gene duplications as a major evolutionary driver of coral biology" is too obvious and plain to me. Gene duplication as the driver of genome evolution has been well established since the first bacterial genomes (even before, it was known but confirmed with hard evidence with three complete bacterial genomes). This reviewer recommend it to be a bit more specific. Such as adding duplicated gene families. "coral biology" is too general. If it was to do with some kind of resistance to environmental insults, it can be said at the title level. Also, my guess is that previous genome assembly studies must have shown data supporting it.

Also, it is not really novel or surprising. Is it? Just reading the title, I expected it to be some kind of extensive gene duplication analyses, but the paper is a small book of comparing newly constructed high quality coral genomes with existing ones, with all kinds of typical analyses.

Also, the paper is about "tandemly" duplicated genes rather than gene duplication itself.

We agree with the reviewer that gene duplication has already been established as a driver of genome evolution. We modified the title and tried to be more specific : "Pervasive tandem duplications and convergent evolution shape coral genomes".

Abstract:

2. "a relatively higher gene number" in abstract. It will be good to denote the actual numbers. (forcing readers to sift through numbers... while it is the foundation of the research problem).

We modified the text accordingly and added the number of genes.

3. "a high number of tandemly duplicated genes, ". Again. This is a genome paper and I, as a reader, want to know how many such TDGs are in the coral assemblies as soon as possible. That is the point of abstract anyway.

We added the proportion of TDGs in the abstract.

4. "which are generally difficult to assemble and annotate using short-read sequencing technologies". This is not necessary in abstract. Wasting the word count...

We deleted this information.

5. "These duplicated genes, " At least some gene families will make readers to know what "these" really look like.

We changed the wording.

Introduction:

7. "We posit that gene duplication is an evolutionary driver of coral biology that contributes to their longevity"

 Gene duplication is always the driver of genome evolution. That is why we have almost 1Gbp coral genomes now. I.e., it is too obvious. It is not a discovery. In this case, the gene duplication of certain families is the point, IMO. Many previous studies have checked and shown gene expansion and contraction. If there is any novel TDG(s) or TDG mechanism in corals, it would be interesting. I suggest it to be removed or merged with the previous already very long sentence: "Our results expose the vast presence of duplicated gene families in both coral genomes mapping to functions associated with the innate immune system, which escaped previous analyses based on fragmented and incomplete genomes assemblies due to sequencing method constraints."

We have modified the text accordingly : "We posit that these tandem duplications shape current coral genomes and contribute to the longevity of these organisms".

8. "The *P. lobata* and *P. meandrina* genomes were sequenced using a combination of ONT long reads and Illumina short reads"

 No. Rephrase it. The two genomes were not sequenced by a combination of ONT and Ill... reads. By ONT and Illumina "sequencers", "machines" or "platforms".

We replaced "sequenced" by "generated".

9. "and a high level of heterozygosity was detected, 2.3% and 1.14% respectively"

 2.3% is high for a coral? Corals and marine species like that contain quite high heterozygosity as far as I know. High compared to what?

We computed the heterozygosity rate (using Genomescope2) for other sequenced corals (based on short reads), and the rate ranged from 1.1% to 2.2%.

Pocillopora verrucosa 1.11179

Montipora capitata 1.21968

Orbicella faveolata 1.22082

Goniastrea aspera 1.32102

Pocillopora acuta 1.36121

Stylophora pistillata 1.3951

Fungia spp 1.5966

Galaxea fascicularis 1.71077

Acropora millepora 1.84267

Porites lutea 1.99808

Acropora digitifera 2.19456

10. "The resulting haploid assemblies, with a cumulative size of 646 Mb for *P. lobata* and 347 Mb for *P. meandrina*, contained...."

 with cumulative size"s" of *P. lobata* and ... *P. meandrina*...

We modified the text accordingly.

11. "we sequenced the genome of *Porites evermanni* using short-reads."

 Again, you did not sequence it with short-reads. You sequenced it with short-read Illumina sequencers.

We added "technology" : we sequenced the genome of *Porites evermanni* using short-reads technology.

12 "Although more fragmented,"

 Although much fragmented? Or Although more fragmented than XXX, ..

We modified the text accordingly.

13. "These particular genes"

 Particular? It would be clear to say These TDGs...

We replaced "particular" by "duplicated".

14. "the annotation of these TDG"

 I wish those TDGs were stated as early as possible. The paragraph is depicting what the authors have done. It would be better to put forward the results first.

The contribution is such a new assembly protocol that successfully found TDGs and telling what the TDGs as early as possible will make readers to keep paying attention.

(Anyway, this is not critical)

We hesitated also, but we preferred to show global comparisons before describing in more detail TDGs and amplified gene families.

15. "Noteworthy, we noticed in the three Porites species"

 Rephrase this. Noteworthy.. noticed...

We replaced Noteworthy by Strikingly.

16. "Attempts to search for this sequence failed to find it outside the Porites genus"

 It is not natural. Search for.... failed and again to "find it". Just "Attempts to search for this sequence failed outside the Porites genus" will be fine.

We modified the text accordingly.

17. "true duplications from allelic copies of a given gene."

 ...duplicates?

We modified the text accordingly.

18. "To further compare coral genomes, orthologs with 22 other cnidarian species were calculated"

 What do you mean by "calculated"? analyzed?

We replaced "orthologs with 22 other cnidarian species were calculated" by "orthologous relationships within 25 cnidarian species were identified".

19. "Notably, although an initial matter of debate,"

 Notably, as an initial matter/point for debate? Rephrasing will be good.

We modified the text accordingly.

Reviewer #3:

The manuscript by Noel et al. 2022 focuses on the generation and analysis of two new coral genomes (*Porites lobata* and *Pocillopora meandrina*) sequenced with Oxford Nanopore technology. After performing gene prediction on the assemblies, the authors found a significantly higher number of predicted genes in these genomes than in other coral (and other non-coral cnidarian) genomes and explain this by identifying a high number of tandemly duplicated genes and make claims about why certain gene families may have been duplicated in these lineages.

Although the authors have done an admirable job in generating these important genome assemblies and made an attempt to try to detect allelic duplications in their assemblies and perform haplotype phasing using Haplomerger2 to generate haploid versions of their assemblies, I am not fully convinced that this process has successfully phased their assemblies and fully removed "false duplications" from their haploid assemblies. Unfortunately, if false duplications remain, it can negatively affect all downstream analyses. Therefore, interpretations of the data and the evidence supporting the main findings of this study that there are a high number of tandemly duplicated genes in two coral genomes may be weak.

I give my reasoning and evidence to support this line of thinking below.

As noted in Guan et al. 2020 (Bioinformatics 36(9) 2896- 28980), several tools have been developed to try to resolve the problem of "false duplication" in genome assemblies and Haplomerger2 is one such tool. The main problem with Haplomerger2 - as pointed out in this paper - is that it ignores read depth and relies only on the alignment of contigs to each other. The authors of that paper suggest that Haplomerger2 can make false joins to link adjacent contigs leading to false duplications, among other issues, and suggest trying their tool, "purge_dups", which does account for read depth, to overcome these issues.

Additionally, a preprint by Ko et al. 2021 (<https://doi.org/10.1101/2021.04.09.438957>) goes into great depth on the topic of how false duplications in genome assemblies lead to false biological conclusions. Falsely duplicated genomic regions and genes can lead to overestimated gene family expansions. The main source of this occurs where haplotype sequences are more divergent than other parts of the genome which can lead assembly algorithms to classify them as separate regions or genes. Ko et al. discuss how better methods are still needed to better separate haplotypes and urge extremely cautious analysis on gene gains. Without proper phasing/haplotype separation, both alleles of heterozygous loci can be assembled into one scaffold (or two different scaffolds). Ko et al. use two independent methods to try to tackle these issues, including Cactus for whole-genome alignment between two assemblies (they were comparing different assemblies) and "purge_dups" for self-alignment.

I agree with the points made by Guan et al. and Ko et al. and urge Noel et al. to be extremely cautious in their interpretation of their current assemblies. I believe that more testing and careful analysis needs to be done to determine if they still have "false duplications" present in their haploid assemblies before moving forward with their analyses of the genomes. My main reasons for concern stem from three pieces of information presented in the manuscript (1) these corals have very high estimated levels of heterozygosity across their genomes (*P. lobata*

estimated at 2.3% and *P. meandrina* estimated at 1.14%), (2) the fact that their haploid assembly sizes are significantly larger than their estimated genome sizes (*P. lobata* estimated genome size 543 Mb, final assembly size is 646 Mb; *P. meandrina* estimated genome size 315 Mb, final assembly size is 347 Mb) and (3) their predicted gene numbers are much higher than other cnidarian genomes (*P. lobata* has 42,872 predicted genes, *P. meandrina* has 32,095 predicted genes).

To me, these three statistics amount to red flags that provide ample evidence that strongly indicates that the genome assembler they chose (SMARTdenovo) as well as their Haplomerger2 analysis were unable to completely resolve the two haplotypes and they have retained some false duplications in their haploid assemblies, especially in *P. lobata*. The authors use BUSCO and KAT analysis to show a reduction in allelic duplications, but this reduction does not appear to be "complete" which could lead to false duplications remaining in some areas of the genome, particularly the regions that appear to contain tandemly duplicated genes. Even the Ks distribution data in Figure 5 for the tandemly duplicated gene pairs seems to support the idea that these could be false duplications, likely retained due to the fact that the phasing was incomplete due to high heterozygosity in these regions.

At the very least, I would urge the authors to run 'purge_dups' on their assemblies and examine how their resulting primary haploid assemblies compare with their current assemblies from Haplomerger2. I would also recommend running some k-mer profiling analyses on the resulting assemblies after running 'purge_dups', like in Extended Data Fig. 2 of Ko et al. 2021. There are also assemblers, like NextDenovo, that have been shown to perform better on more repetitive and highly heterozygous genomes (e.g. Sun et al. 2021 <https://doi.org/10.1098/rstb.2020.0160>) that the authors could try. It will be very important to perform more careful analysis of their haplotype assemblies to identify and remove false duplications before making claims about gene gains and tandemly duplicated genes in these corals.

Since I have these concerns about the underlying assemblies, I will limit my comments to this aspect of the manuscript as it can affect nearly all the analyses and conclusions that follow. As Ko et al. (2021) point out, even though long-read sequencing is better at resolving repetitive regions of genomes, long reads are not able to fully resolve false duplications and I suspect that this manuscript may suffer from a significant proportion of false allelic duplications remaining in their assemblies, obscuring the authors' ability to distinguish between true duplications and false duplications and to make well-supported biological conclusions.

I do think these coral genomes make an important contribution to cnidarian genomics, and in particular, coral genomics, and I found the overall writing and methods used in general to be clear and reasonable. This problem of false allelic duplication is a very tricky and complicated one that has not yet been solved, particularly for genomes with high levels of heterozygosity - even for vertebrate genomes - and I hope the authors will not take my comments as a condemnation of their science but rather an attempt to provide important up-to-date information on this very real problem and an opportunity to use additional tools to better assess if what they are seeing with the tandemly duplicated genes is actually "real" or not. I would be happy to take another look at the manuscript after the authors perform additional analyses and provide more evidence that they have gone further to address the problem of false duplication in their data.

We thank the reviewer for their comment and understand their doubts regarding the veracity of the identified duplicated genes. At the time we started *de novo* assembly of our coral genomes, the Purge Dups tool had not yet been released. We chose haplomerger2 because it shows excellent results on a large number of heterozygous organisms. However, in light of the two cited articles (Guan et al. and Ko et al.), we understand that strong arguments need to be added to convince readers that the many duplicated genes found in coral genomes are not the results of biases in assemblies.

For this purpose, we first validated the methodological choice we made, by comparing the Purge Dups tool with our approach based on haplomerger2, and as recommended by the reviewer, we performed kmer profiling of the resulting assemblies. Second, and maybe more importantly, we validated the structure of duplicated gene clusters using Nanopore long reads. In addition, we performed a draft gene prediction on the alternative haplotype of each assembly and compared the substitution rate between pairs of allelic and tandemly duplicated genes.

Briefly, the assembly obtained using Purge Dups for *Porites lobata* is very similar to the one obtained using Haplomerger2 (Table S3 and Figure S4) which is a confirmation of the great work performed by Haplomerger2. Surprisingly, Purge Dups produced good results on *Porites lobata* genome assembly, but did not alter the raw assembly of *Pocillopora meandrina*, despite adequate coverage thresholds. This may be due to the lower proportion of haplotypic duplications (Figure S4). More importantly, a high proportion of pairs of tandemly duplicated gene (70% for *Porites lobata* and 91% for *Pocillopora meandrina*) and TDGs clusters (45% for *Porites lobata* and 67% for *Pocillopora meandrina*) are validated by long reads which is a strong support to our findings about duplicated genes in corals (Table S9 and Figures S6, S7, S8 and S9). Additionally, and not surprisingly, allelic pairs and true duplicated genes have very different conservation patterns that can help discriminate false duplications (allelic copies) from true amplified genes (Figure 5B).

These analyses led to new paragraphs in the main text and methods section, and obviously reinforced the methodologies used in our work. We hope that these additional results will help the reviewer and readers to be convinced of the presence of pervasive duplications at the gene level in coral genomes.

Second round of review

Reviewer 3

I appreciate the authors taking my suggestion to run the `purge_dups` tool and comparing the output assemblies with the Haplomerger2 assemblies, including by performing kmer profiling.

I also appreciate that the authors have now validated the structure of duplicated gene clusters using Nanopore long reads.

These two additional steps and the added sections of the manuscript have strengthened the claims and have made me more confident that the duplicated gene regions exist.

However, I am still not fully convinced that proper tools exist to completely separate the two allelic versions and avoid the problem of false allelic duplication. I am not convinced that the two allelic versions were completely separated based on the evidence provided. It is clear that the allelic duplications are significantly reduced by the KAT plots and BUSCO duplicate percentages, but they are not completely eliminated. Importantly, the Nanopore long reads validated some but not all pairs of adjacent duplicated reads or clusters (Table S9).

Thus, the sentence added to the section "Comparison with available genomes" is too strongly worded.

Original sentence added:

"In our assemblies of *P. lobata* and *P. meandrina*, BUSCO and KAT analyses showed a reduction of the allelic duplications which confirms that the two allelic versions were successfully separated (Tables S3 and S5 and Figures 1G, S2, S3 and S4)."

I suggest changing it to:

"In our assemblies of *P. lobata* and *P. meandrina*, BUSCO and KAT analyses showed a reduction of the allelic duplications which suggests that the two allelic versions were successfully separated as much as possible with currently available tools (Tables S3 and S5 and Figures 1G, S2, S3 and S4)."

Also for the BUSCO results, each time a result is reported, please state which lineage dataset was used.

In the methods, the metazoa Odb10 dataset of 954 is mentioned. But I can't seem to find which dataset was used for BUSCO for Table S3. I only see "BUSCO (N=255)".

Is this the eukaryota dataset? Why use the metazoa dataset (954) in some places but the eukaryota dataset in others? Please clarify in the Table S3 legend.

The remainder of the manuscript gives additional analyses and all of these seemed reasonable and clear.

If the authors made the few changes I suggest, I am satisfied with this manuscript being published.

Reviewer #3:

I appreciate the authors taking my suggestion to run the `purge_dups` tool and comparing the output assemblies with the Haplomerger2 assemblies, including by performing kmer profiling. I also appreciate that the authors have now validated the structure of duplicated gene clusters using Nanopore long reads. These two additional steps and the added sections of the manuscript have strengthened the claims and have made me more confident that the duplicated gene regions exist.

However, I am still not fully convinced that proper tools exist to completely separate the two allelic versions and avoid the problem of false allelic duplication. I am not convinced that the two allelic versions were completely separated based on the evidence provided. It is clear that the allelic duplications are significantly reduced by the KAT plots and BUSCO duplicate percentages, but they are not completely eliminated. Importantly, the Nanopore long reads validated some but not all pairs of adjacent duplicated reads or clusters (Table S9). Thus, the sentence added to the section "Comparison with available genomes" is too strongly worded.

Original sentence added:

"In our assemblies of *P. lobata* and *P. meandrina*, BUSCO and KAT analyses showed a reduction of the allelic duplications which confirms that the two allelic versions were successfully separated."

I suggest changing it to:

"In our assemblies of *P. lobata* and *P. meandrina*, BUSCO and KAT analyses showed a reduction of the allelic duplications which suggests that the two allelic versions were successfully separated as much as possible with currently available tools."

We agree with the reviewer and change the text as suggested.

Also for the BUSCO results, each time a result is reported, please state which lineage dataset was used. In the methods, the metazoa odb10 dataset of 954 is mentioned. But I can't seem to find which dataset was used for BUSCO for Table S3. I only see "BUSCO (N=255)". Is this the eukaryota dataset? Why use the metazoa dataset (954) in some places but the eukaryota dataset in others? Please clarify in the Table S3 legend.

We agree with the reviewer and have explicitly added in Table S3 the dataset used and not just the number of genes.

The remainder of the manuscript gives additional analyses and all of these seemed reasonable and clear. If the authors made the few changes I suggest, I am satisfied with this manuscript being published.

We thank the reviewer for their comments and feedback on this new version of our article.